# mRNA poly(A)-tail changes specified by deadenylation broadly reshape translation in Drosophila oocytes and early embryos

Stephen W Eichhorn[1,2], Alexander O Subtelny[1,2,3], Iva Kronja[2,4], Jamie C Kwasnieski[1,2], Terry L Orr-Weaver[2,4*], David P Bartel[1,2*]

[1]Howard Hughes Medical Institute, Whitehead Institute for Biomedical Research, Cambridge, United States; [2]Department of Biology, Massachusetts Institute of Technology, Cambridge, United States; [3]Harvard-MIT Division of Health Sciences and Technology, Cambridge, United States; [4]Whitehead Institute for Biomedical Research, Cambridge, United States

**Abstract** Because maturing oocytes and early embryos lack appreciable transcription, posttranscriptional regulatory processes control their development. To better understand this control, we profiled translational efficiencies and poly(A)-tail lengths throughout Drosophila oocyte maturation and early embryonic development. The correspondence between translational-efficiency changes and tail-length changes indicated that tail-length changes broadly regulate translation until gastrulation, when this coupling disappears. During egg activation, relative changes in poly(A)-tail length, and thus translational efficiency, were largely retained in the absence of cytoplasmic polyadenylation, which indicated that selective poly(A)-tail shortening primarily specifies these changes. Many translational changes depended on PAN GU and Smaug, and these changes were largely attributable to tail-length changes. Our results also revealed the presence of tail-length–independent mechanisms that maintained translation despite tail-length shortening during oocyte maturation, and prevented essentially all translation of *bicoid* and several other mRNAs before egg activation. In addition to these fundamental insights, our results provide valuable resources for future studies.

*For correspondence: weaver@wi.mit.edu (TLO-W); dbartel@wi.mit.edu (DPB)

**Competing interests:** The authors declare that no competing interests exist.

## Introduction

The generation of a zygote through fertilization marks the onset of development for the offspring, and requires the proper completion of sperm and oocyte development in the respective parents. Spermatogenesis produces haploid sperm capable of penetrating the oocyte, whereas oogenesis produces differentiated oocytes that are stockpiled with maternal nutrients, proteins, and mRNAs, and have outer layers that protect the embryo and enable fertilization. In most animals the oocyte is arrested in meiosis, and fertilization leads to initiation of mitosis as the oocyte nucleus completes meiosis and fuses with the haploid sperm nucleus. Fertilization also leads to changes in mRNA translation and protein stability, which support a period of development driven off of maternal stockpiles. After this initial stage of maternal control, which lasts for 1–2 mitotic divisions in mammals and 13 mitotic divisions in Drosophila, widespread transcription begins from the zygotic nuclei (*Tadros and Lipshitz, 2009*).

The production of viable offspring requires three key developmental events: oocyte maturation, the oocyte-to-embryo transition (OET), and the maternal-to-zygotic transition (MZT) (*Figure 1A*).

**Figure 1.** Transient coupling between poly(A)-tail length and translational efficiency in embryos. (**A**) Time line of the developmental transitions probed in this study, illustrating the presence of maternal and zygotic mRNAs (pink and purple, respectively). (**B**) Relationship between mean poly(A)-tail length and relative translational efficiency (TE) at the indicated developmental stage (*top*), and the relationship between tail-length and TE changes observed between adjacent stages (*bottom*). For each stage, tail-length and TE measurements were from the same sample. In the top panels, results are plotted for all mRNAs that had ≥100 poly(A)-tail length measurements (tags), ≥10.0 reads per million mapped reads (RPM) in the RNA-seq data, and >0 RPM in the ribosome-profiling data. In each sample, TE values (log2) were median centered by subtracting the median TE value of that sample from each value (median values in stage 14 oocytes, 0–1 hr embryos, 2–3 hr embryos, 3–4 hr embryos, and 5–6 hr embryos were –0.2523, –0.1329, 0.4799, 0.6469, and 0.6447, respectively). In the bottom panels, results are plotted for all mRNAs that had ≥100 poly(A) tags and ≥10.0 RPM in the RNA-seq data of both samples, and ≥10.0 RPM in the ribosome-profiling data of one of the two samples and >0 RPM in the other. TE fold-change values (log2) were median centered for each comparison (median values, 0.2027, –0.0047, 0.0701, and –0.003 from left to right). The Spearman correlation coefficient ($R_s$) and number of mRNAs (*n*) are shown in each plot. The TE data for stage 14 oocytes were from *Kronja et al. (2014b)*. (**C**) Abundance and TE of *zelda*, *stat92E*, *smaug*, and *bicoid* mRNA at the indicated developmental stages (RPKM is reads per kilobase per million mapped reads). TE values (log2) for each stage were median centered as described above, using the median values (log2) reported in the legends of *Figure 1B* and *Figure 2*. The data for stage 14 were from *Kronja et al. (2014b)*. If no RPF reads were observed in a sample, TEs were calculated using a pseudocount of 1 read. Points for

*Figure 1 continued on next page*

*Figure 1 continued*

which TE is based on either a single read or pseudocount are in grey, as are the lines connecting to them. Other TEs were based on ≥25 RPF reads, except the TEs for *bicoid* in stage 14 oocytes and 3–4 hr embryos (3 and 9 reads, respectively) and the TE for *smaug* in stage 13 oocytes (11 reads).

The following figure supplement is available for figure 1:

**Figure supplement 1.** Dynamics of translational regulation during the OET and early embryonic development.

Oocyte maturation involves the release of the primary meiotic arrest at prophase I and progression of the oocyte nucleus into meiotic divisions to produce a mature oocyte (egg) capable of being fertilized (*Von Stetina and Orr-Weaver, 2011*). The mature oocyte is arrested in metaphase II in most vertebrates and metaphase I in insects. The OET requires egg activation, which is coupled to fertilization in most animals. Egg activation releases the secondary meiotic arrest, allowing completion of meiosis (*Horner and Wolfner, 2008*). The MZT marks the transfer of control of development from the mother to the zygote as maternal mRNAs are degraded, transcription from the zygotic genome begins, and embryonic development becomes dependent on zygotic gene products (*Tadros and Lipshitz, 2009*). In Xenopus, zebrafish, and Drosophila the major activation of zygotic transcription occurs as the cell cycle lengthens and gastrulation begins, a developmental period referred to as the midblastula transition.

In nearly all animals, oocyte maturation and the OET occur in the absence of transcription, and thus any changes in protein levels must occur through posttranscriptional regulatory mechanisms. Indeed, widespread changes in protein levels without corresponding changes in mRNA levels are observed at these developmental transitions (*Kronja et al., 2014a*, *2014b*). Posttranscriptional regulation of gene expression during oocyte maturation, the OET, and early embryogenesis faces unique challenges, including stably storing maternal mRNAs through the prolonged periods of oogenesis and maintaining some of these mRNAs in a translationally inactive state (referred to as masking) while selectively translating others at the proper times in development. During mouse oocyte maturation ~1300 mRNAs are translationally upregulated and ~1500 are translationally downregulated (*Chen et al., 2011*). Similarly, during Drosophila egg activation ~1000 mRNAs are translationally upregulated, and ~500 are translationally repressed (*Kronja et al., 2014b*).

One long-appreciated mechanism of translational regulation during development acts through control of mRNA poly(A)-tail length (*Richter, 2007*; *Weill et al., 2012*). The observation of short poly(A) tails in oocytes from amphibians and marine invertebrates led to the proposal that short poly(A) tails help mask maternal mRNAs, preventing translation without triggering the destabilization that typically results from deadenylation. Specific mRNAs are then targeted for cytoplasmic polyadenylation, and the lengthened poly(A) tails in turn cause translational upregulation of these mRNAs (*McGrew et al., 1989*), as demonstrated for the mRNAs for *c-mos* and several cyclin genes during oocyte maturation in Xenopus (*Sheets et al., 1994*; *Barkoff et al., 1998*). The Cytoplasmic Polyadenylation Element Binding Protein (CPEB) plays a critical role in controlling translation of maternal mRNAs, both in recruiting Maskin to repress translation and later in promoting polyadenylation by binding cytoplasmic poly(A) polymerase together with Cleavage and Polyadenylation Specificity Factor (CPSF) (*Richter, 2007*; *Weill et al., 2012*; *Ivshina et al., 2014*). In Drosophila, the cytoplasmic poly(A) polymerase encoded by the *wispy* gene promotes poly(A)-tail lengthening during both oocyte maturation and egg activation (*Benoit et al., 2008*; *Cui et al., 2008*, *2013*), and increased poly(A)-tail lengths of mRNAs for *cyclin B, cortex, bicoid, torso*, and *Toll* coincide with the translational upregulation of these mRNAs (*Salles et al., 1994*; *Lieberfarb et al., 1996*; *Vardy and Orr-Weaver, 2007a*; *Benoit et al., 2008*). Moreover, appending synthetic poly(A) tails of increasing lengths to the *bicoid* mRNA promotes the translation of this mRNA, which demonstrates a causal relationship between increased poly(A)-tail length and increased translation in early Drosophila embryos (*Salles et al., 1994*).

Translational repressors, some of which have been identified through genetic analysis of pattern formation in the Drosophila embryo, also act during oogenesis and early embryogenesis (*Vardy and Orr-Weaver, 2007b*; *Besse and Ephrussi, 2008*; *Barckmann and Simonelig, 2013*). These repressors can impact either poly(A)-tail length (*Temme et al., 2014*), the translational machinery, or both. Brain Tumor and Pumilio (PUM) bind to target mRNAs and recruit the CCR4-NOT deadenylase

complex, which shortens poly(A) tails (*Kadyrova et al., 2007*; *Temme et al., 2010*; *Newton et al., 2015*). Bruno binds to target mRNAs during oogenesis and recruits the eIF4E-binding protein Cup, which represses translation by both blocking the interaction between eIF4E and eIF4G (*Nakamura et al., 2004*) and promoting deadenylation while repressing decapping, which traps mRNAs in a translationally repressed state (*Igreja and Izaurralde, 2011*). Similarly, the RNA-binding protein Smaug (SMG) is reported to recruit either Cup or the CCR4-NOT deadenylase complex, with the consequent repression of translation (*Nelson et al., 2004*; *Semotok et al., 2005*). *smg* mRNA is translationally repressed in the mature oocyte by PUM, and at egg activation this repression is relieved by the PAN GU (PNG) protein kinase complex (*Tadros et al., 2007*). PNG is also thought to regulate other translational regulators.

mRNA expression arrays, RNA-seq and ribosome-footprint profiling have enabled global analyses of mRNA levels and relative translational efficiencies (TEs) in Drosophila oocytes and embryos (*Qin et al., 2007*; *Dunn et al., 2013*; *Chen et al., 2014*; *Kronja et al., 2014b*). Measuring poly(A)-tail lengths has been more challenging due to the homopolymeric nature of the tail, and this technical hurdle has restricted studies to measuring tail lengths of a small number of Drosophila mRNAs (*Salles et al., 1994*; *Lieberfarb et al., 1996*) or to estimating relative differences in tail lengths more broadly using hybridization-based methods (*Cui et al., 2013*). However, methods are now available to measure poly(A)-tails of essentially any length for millions of individual mRNA molecules (*Chang et al., 2014*; *Subtelny et al., 2014*). In zebrafish and Xenopus, high-throughput measurements demonstrate that poly(A)-tail length and TE are globally coupled during early embryonic development (*Subtelny et al., 2014*), as expected from single-gene studies (*McGrew et al., 1989*; *Sheets et al., 1994*; *Lieberfarb et al., 1996*; *Barkoff et al., 1998*; *Pesin and Orr-Weaver, 2007*; *Benoit et al., 2008*). Surprisingly, however, this coupling disappears at gastrulation and is not observed in any of the post-embryonic samples examined (*Subtelny et al., 2014*). Thus, soon after a developmental shift in transcriptional control (the MZT), these vertebrate species undergo a developmental shift in translational control, in which TE abruptly changes from being strongly coupled to tail length to being uncoupled.

*Drosophila melanogaster* affords the opportunity to define changes in poly(A)-tail length and TE during oocyte maturation, the OET, and the MZT, facilitated by the high quality of both the genome assembly and the coding-sequence (CDS) annotations, the wealth of knowledge on regulatory proteins active during early development, the availability of genetic mutants in these known regulatory proteins, and the ability to isolate sufficient quantities of oocytes and embryos at distinct developmental stages for high-throughput sequencing techniques. Here we systematically analyze changes in poly(A)-tail length and TE during oocyte maturation, the OET and the MZT, and delineate the roles of Wispy, PNG, and SMG in regulating translation during these crucial periods. These studies reveal fundamental insights into the regulatory mechanisms and regimes operating during these key developmental transitions.

## Results

### A conserved switch in the nature of translational control

We performed poly(A)-tail length profiling by sequencing (PAL-seq) (*Subtelny et al., 2014*) to measure poly(A)-tail lengths globally at eight developmental stages in *Drosophila melanogaster*, beginning with oocytes at the primary meiotic arrest point and ending with embryos late in gastrulation, at germ-band retraction (*Figure 1A*). PAL-seq determines the poly(A)-tail length of millions of mRNA molecules, providing for each molecule both a fluorescent measurement that reports on the poly(A)-tail length and a sequencing read that reports the cleavage and poly(A) site. Because alternative start sites or alternative splicing can generate different transcripts with the same poly(A) site, we considered our results with respect to a curated set of unique gene models. Moreover, because alternative cleavage and polyadenylation can generate different poly(A) sites for mRNAs from the same gene, we used our PAL-seq data to identify for each gene the alternative isoform with the longest 3′ UTR and then combined its tail-length measurements with those of any shorter tandem 3′-UTR isoforms. Thus, the 'mRNAs' of our analyses each corresponded to a unique gene model representing a different protein-coding gene and the aggregated tail-length results from many (if not all) of the alternative transcripts from that gene. On the same samples as those analyzed by PAL-

seq, we performed ribosome-footprint profiling and RNA-seq to determine TEs (*Ingolia et al., 2009*), which also report aggregate results for alternative isoforms, allowing comparison of mean poly(A)-tail length and TE for the thousands of mRNAs that passed our quantification cutoffs at each stage.

In Drosophila, the first 13 division cycles occur in a nuclear syncytium. Cellularization (at 2–3 hr of embryogenesis) marks completion of the MZT and is immediately followed by the morphogenetic movements of gastrulation. Thus, early embryogenesis is controlled by maternal mRNAs, and based on the studies of mRNAs from a few genes (*Salles et al., 1994*; *Lieberfarb et al., 1996*), those with longer poly(A)-tails would be predicted to have higher TEs, as observed in vertebrate systems (*Subtelny et al., 2014*). Indeed, poly(A)-tail length and TE were correlated in cleavage-stage embryos (0–1 hr embryo), and the slope of this relationship was steep, such that small differences in poly(A)-tail length corresponded to large differences in TE (*Figure 1B*; e.g., the median TE of mRNAs with a poly(A)-tail length between 70–80 nt was >5 fold greater than that of mRNAs with a poly(A)-tail length between 30–40 nt). Likewise, when progressing through the OET, the changes in TE correlated with the changes in tail length (*Figure 1B*, 0–1 hr embryo / Stage 14 oocyte).

Although a positive correlation between poly(A)-tail length and TE persisted in the 2–3 hr embryos, the slope of this relationship strongly diminished such that the median TE of mRNAs with a poly(A)-tail length between 70–80 nt was <2 fold greater than that of mRNAs with a poly(A)-tail length between 30–40 nt, as expected if poly(A)-tail length changes were no longer used to influence translation in these cellularized and gastrulating embryos. Indeed, the changes in poly(A)-tail length that occurred between cleavage-stage and cellularized embryos were not positively correlated with the changes in TE (*Figure 1B*, 2–3 hr embryo / 0–1 hr embryo). The positive correlation between poly(A)-tail length and TE disappeared in subsequent stages of gastrulation (3–4 hr embryos and 5–6 hr embryos), and although tail lengths changed between these stages, the TE changes spanned a very narrow range (*Figure 1B*, 3–4 hr embryo / 2–3 hr embryo and 5–6 hr embryo / 3–4 hr embryo). The tail-length changes observed between these stages presumably resulted from transcription and mRNA decay as well as deadenylation, whereas those in earlier stages resulted primarily from cytoplasmic polyadenylation and deadenylation.

The narrower range of TE values and TE changes following the onset of gastrulation (*Figure 1B*) was consistent with a diminished role of translational regulation in specifying protein output after the embryo begins to modulate gene expression by altering mRNA abundance. Most importantly, these results showed that the developmental shift in the nature of translational control observed in vertebrates also occurs in flies; in Drosophila, just as in zebrafish and Xenopus (*Subtelny et al., 2014*), the coupling between poly(A)-tail length and TE observed in the early embryo disappeared around gastrulation.

An analysis of the temporal patterns of translational regulation occurring from the OET through the onset of gastrulation identified several significantly enriched patterns, although all possible patterns were observed (*Figure 1—figure supplement 1*). Among the five enriched profiles were those for mRNAs that were translationally upregulated at the OET and then stayed elevated during early embryonic development, those for mRNAs that were upregulated at the OET and then downregulated during early embryonic development, and those for mRNAs that were downregulated before the OET and remained repressed (*Figure 1—figure supplement 1*). Because ribosome-profiling data reflect relative rather than absolute levels of translation of an mRNA across developmental stages, these results should be interpreted as reflecting changes in how well an mRNA was translated relative to all other mRNAs in the same stage. Thus, an mRNA translated at the same absolute efficiency in two stages might be perceived as downregulated if translation of many other mRNAs increased. Nonetheless, these dynamic and substantial changes in TE observed during the OET and early embryogenesis revealed a large-scale and rapid change in the translational capacity of mRNAs relative to each other during early development.

Pairwise comparisons within the mRNA, translation, and tail-length datasets (from RNA-seq, ribosome-footprint profiling, and PAL-seq, respectively) for the different developmental stages further supported these overall conclusions (*Supplementary file 1*). As expected, mRNA abundances were highly correlated between all stages prior to the onset of transcription (*Supplementary file 1*). Over these same stages, poly(A)-tail lengths for many different mRNAs shortened or lengthened, but these changes seemed to have minimal impact on mRNA abundance measurements, consistent with stability of shorter-tail mRNAs during this time (and also indicating that the poly(A)-selection that

was performed as part of the RNA-seq protocol did not bias our RNA abundance, and thus TE, measurements). In contrast to the highly correlated mRNA abundances, the number of ribosome-protected fragments (RPFs) differed substantially between stages prior to the onset of transcription (*Supplementary file 1*), consistent with dynamic and robust translational regulation during this time. Once transcription began, widespread changes in both mRNA levels and RPFs were observed between stages (*Supplementary file 1*). Poly(A)-tail lengths varied both prior to and after the onset of transcription, and generally mirrored the pattern of correlations observed for RPFs up until the MZT (*Supplementary file 1*). Little to no positive correlation between poly(A)-tail lengths was observed when comparing samples with no active transcription to those after the onset of robust transcription (*Supplementary file 1*), presumably because of the influence of nascent transcripts on mean poly(A)-tail lengths.

The striking increases in TE observed soon after fertilization presumably generated factors needed for later developmental transitions. For example, the TE of mRNAs for *zelda* and *stat92E*, two transcription factors required for the onset of zygotic transcription at the MZT (*Liang et al., 2008*; *Harrison and Eisen, 2015*), increased markedly during the OET (*Figure 1C*), as expected if translational activation of these mRNAs produces the proteins necessary to initiate the MZT, and accounting for the previously noted increase in Zelda protein in early embryos (*Harrison and Eisen, 2015*). Likewise, the TE of *smaug* and *bicoid* mRNA increased markedly during the OET (*Figure 1C*), yielding factors essential for embryonic development and patterning, respectively. Although the TE of *stat92E* and *smaug* began to increase at stage 14, their TEs were still relatively low until after the OET (*Figure 1C*), consistent with the roles of their translation products during early embryonic development.

## Translational regulation during oocyte maturation

Having established that Drosophila embryos resemble those of fish and frog with respect to the coupling between TE and poly(A)-tail length, we sought to extend these findings to oocytes. The ability to isolate individual oocytes at distinct developmental stages in Drosophila enabled us to measure poly(A)-tail lengths and TEs and examine their relationship at key stages of oogenesis. Stage 11 oocytes are arrested in prophase I. Nuclear envelope breakdown, a hallmark of the release of this arrest, occurs between stages 12 and 13 (*Von Stetina et al., 2008*; *Hughes et al., 2009*), and the oocytes arrest in metaphase I during stage 14, the final stage of oogenesis. Thus, comparison of oocytes from stages 11 through 14 captures the events of meiotic progression and oocyte differentiation that occur over the course of oocyte maturation.

Poly(A)-tail lengths and TEs were well correlated at stages 11, 12, and 13 ($R_s \geq 0.51$), but poorly correlated at stage 14 ($R_s = 0.20$) (*Figure 2*). Previously, the relationship between poly(A)-tail length and translation had been examined for the *c-mos* mRNA and several cyclin mRNAs during Xenopus oocyte maturation (*Sheets et al., 1994*; *Barkoff et al., 1998*; *Piqué et al., 2008*). Our results extended these analyses to an invertebrate and to a global scale, showing that, as for the *c-mos* mRNA and several cyclin mRNAs in the Xenopus oocyte, tail-length increases correspond to translational activation in the maturing Drosophila oocyte. Thus, the global coupling between poly(A)-tail length and TE, which had previously been examined only after fertilization (*Subtelny et al., 2014*), extends well before fertilization to potentially influence key developmental transitions in the oocyte.

Of the mRNAs exceeding our read cutoffs, nearly half (46%) had net TE changes of $\geq 2$ fold over the course of oocyte maturation, with mRNAs for 1264 and 1037 genes up- or downregulated, respectively (*Figure 2—figure supplement 1A*). In contrast, far fewer mRNAs (1.5%) changed in abundance $\geq 2$ fold (25 mRNAs increased, and 52 mRNAs decreased), indicating the posttranscriptional nature of this regulation. More detailed comparison of TEs for each mRNA from stages 11 through 14 revealed unexpectedly dynamic and complex temporal patterns of translational control (*Figure 3A*). The relative TEs for some mRNAs increased progressively at each stage of oogenesis (*Figure 3A* cluster 1), and this set of mRNAs included *cyclin B* (*Figure 3B*), whose translational control via polyadenylation is reported to be crucial for oocyte maturation (*Benoit et al., 2005*). Some of the downregulated mRNAs similarly showed progressive decreases in TE through the stages of oocyte maturation (*Figure 3A* cluster 26). However, more complex patterns of developmental changes in TE were more commonly observed (*Figure 3A*). Moreover, changes in TE correlated poorly with previously measured changes in protein abundance (*Kronja et al., 2014a*), potentially

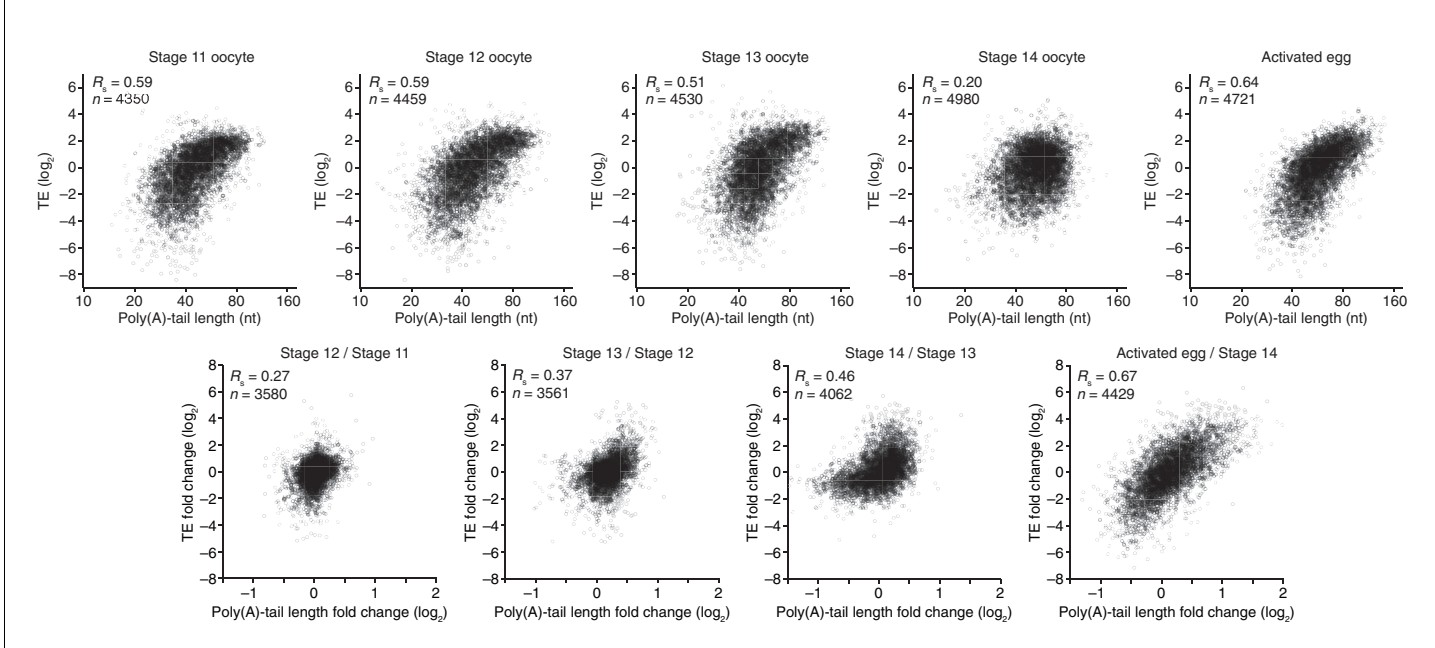

**Figure 2.** Coupling between poly(A)-tail length and translational efficiency in oocytes. Relationship between mean poly(A)-tail length and TE in either oocytes at the indicated stage or activated eggs (*top*), and the relationship between tail-length and TE changes observed between adjacent stages (*bottom*). TE values (log$_2$) were median centered (median values in stage 11, 12, 13, and 14 oocytes and activated eggs were –0.6641, –0.9006, –1.38, –0.2523, and –0.0158, respectively). TE fold-change values (log$_2$) were median centered for each comparison (median values, –0.1961, –0.3969, 1.0886, and 0.2123 from left to right). The TE data for stage 14 oocytes and activated eggs were from *Kronja et al. (2014b)*, and the plot of mean poly(A)-tail length and TE for the stage 14 oocyte is redrawn from *Figure 1B*. Otherwise, these panels are as in *Figure 1B*.

The following figure supplements are available for figure 2:

**Figure supplement 1.** Widespread translational regulation during oocyte maturation.

**Figure supplement 2.** Uncoupled behavior observed for a subset of mRNAs between the last stages of oocyte maturation.

due to a large role of posttranslational regulation during this time (*Figure 2—figure supplement 1B*, $R_s$ = 0.15).

The net changes in TE and poly(A)-tail length over the course of oocyte maturation were largely concordant for both translationally up- and downregulated mRNAs, indicating strong coupling between TE and poly(A)-tail length during oocyte maturation (*Figure 4A*). Nonetheless, a potential deviation from the typical coupled behavior occurred between oocyte stages 13 and 14, when a set of mRNAs that underwent substantial tail-length reductions did not undergo concordant TE reductions (*Figure 2*, Stage 14 / Stage 13, and *Figure 2—figure supplement 2A*). These mRNAs tended to have both long poly(A)-tails and efficient translation in stage 13 oocytes, and then remained well translated compared to other mRNAs in stage 14 oocytes, despite undergoing substantial tail shortening (*Figure 2—figure supplement 2B*). Notable among this set were mRNAs for 22 of the 30 genes encoding subunits of the proteasome (*Figure 2—figure supplement 2A and B*).

To examine the relationship between tail-length changes and TE changes for mRNAs of key factors in oocyte maturation, we focused on those of three of the most highly upregulated cell-cycle regulators involved in meiotic progression, Cyclins B and B3 and the Cdc20 ortholog Fizzy (*Jacobs et al., 1998*; *Swan and Schupbach, 2007*). Between oocyte stages 11 through 14, the TEs of these mRNAs increased 103, 248, and 319 fold (compared to a median TE change of only 1.3 fold), respectively, whereas poly(A)-tail lengths increased 1.6, 2.4, and 2.0 fold (compared to the median of 1.2 fold) (*Figure 4A*). In addition to these examples, mRNAs from 121 other genes were very poorly translated in stage 11 oocytes (TE $\leq$ 0.1) and became $\geq$10-fold better translated during oocyte maturation (*Figure 2—figure supplement 1C*). As with *cyclin B, cyclin B3* and *fizzy* mRNAs,

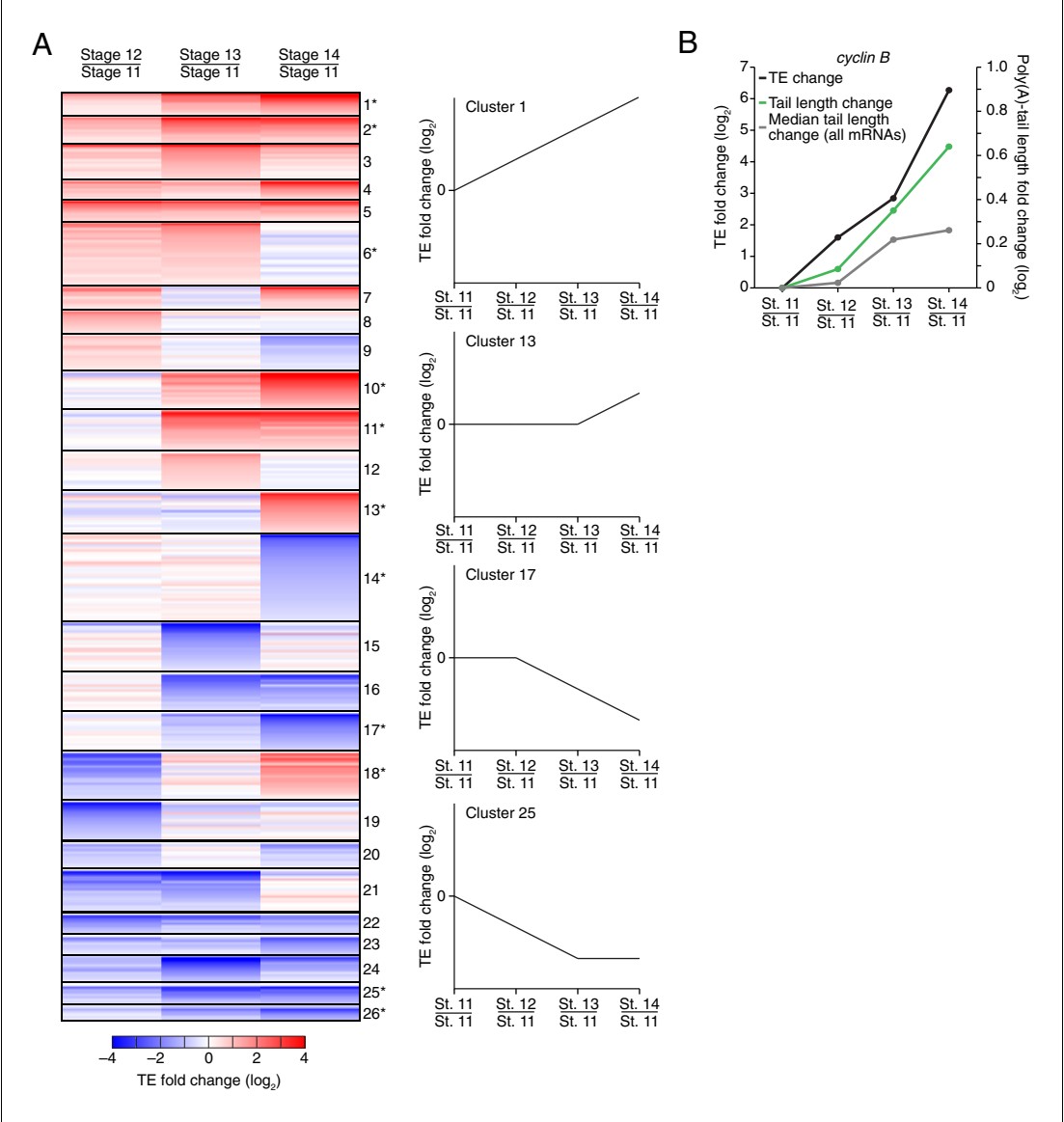

**Figure 3.** Dynamics of translational regulation during oocyte maturation. (A) Distinct patterns of TE changes during oocyte maturation. TE values (log$_2$) for mRNAs in oocytes from stages 11 through 14 that had ≥10.0 RPM in the RNA-seq data for all samples, and ≥10.0 RPM in the ribosome-profiling data for at least one sample and >0.0 RPM in the others were median centered as in *Figure 2*. The median-centered TEs for stage 12, 13, and 14 oocytes were normalized to those of stage 11 oocytes and then clustered into defined patterns. The heatmap shows clustered TE changes for all mRNAs with ≥0.5 or ≤–0.5 log$_2$ fold change between any two samples. Clusters are identified by number, and stylized graphs illustrate the TE dynamics of sample clusters. Clusters that are significantly overrepresented are marked with an asterisk (*p* value < 0.05, following Bonferonni correction). (B) Changes in TE (black) and mean poly(A)-tail length (green) of *cyclin B* mRNA during oocyte maturation. TEs (log$_2$) were normalized as in *Figure 3A*, and the median of the changes in mean poly(A)-tail length (log$_2$) for all mRNAs over these stages is shown as a grey line. Fold changes (log$_2$) calculated relative to stage 11 oocytes are indicated on the respective *y* axes.

these also had modest increases in tail length (*Figure 2—figure supplement 1C*; median tail-length fold change = 1.7 for mRNAs with TE ≤ 0.1 in stage 11 oocytes and a ≥10-fold increase in TE in stage 14 oocytes), raising the question of the degree to which modest tail-length increases might explain the large TE increases observed during oocyte maturation.

One possibility is that a subset of the mRNAs for a gene underwent extensive polyadenylation, and this change in a subset of mRNAs did not cause much change in the mean tail-length value despite causing a large change in TE. However, for *cyclin B*, *cyclin B3* and *fizzy* mRNAs, the

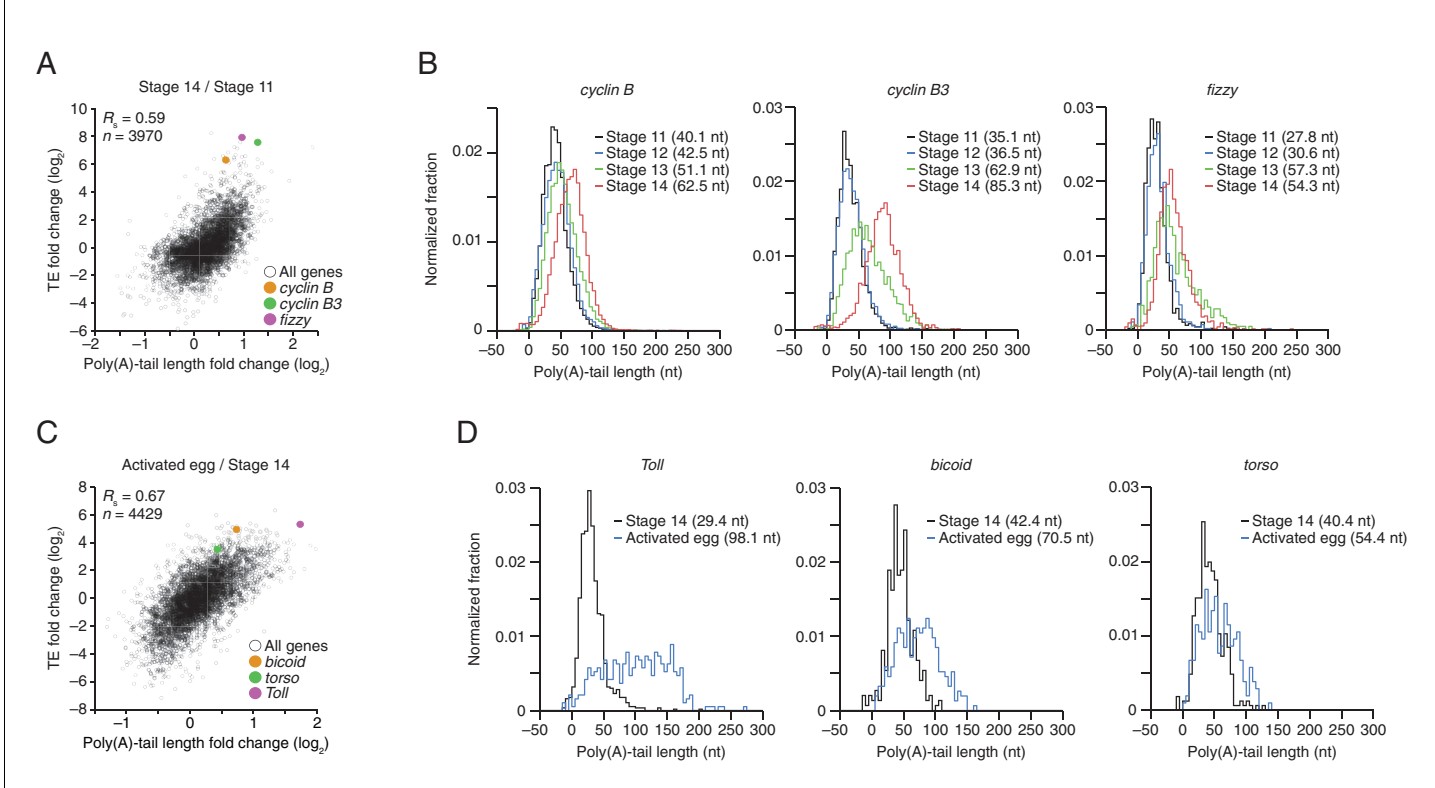

**Figure 4.** TEs and Poly(A) tail lengths of selected mRNAs during oocyte maturation and egg activation. (**A**) Relationship between net tail-length and TE changes observed after oocyte maturation, highlighting the behavior of *cyclin B, cyclin B3*, and *fizzy*. The TE data for stage 14 oocytes were from *Kronja et al. (2014b)*. Values from stage 14 oocytes are compared to those of stage 11 oocytes (median TE value used for median centering, 0.389); otherwise, as in *Figure 1B*. (**B**) Distributions of poly(A)-tail lengths of mRNAs from *cyclin B (left), cyclin B3 (middle)*, and *fizzy (right)* at the indicated developmental stages. Mean poly(A)-tail lengths are in parentheses. (**C**) Relationship between tail-length and TE changes observed after egg activation, showing the same plot as in *Figure 2* (bottom right), but highlighting the behavior of *Toll, bicoid*, and *torso*. (**D**) Distributions of poly(A)-tail lengths of mRNAs from *Toll (left), bicoid (middle)*, and *torso (right)* at the indicated developmental stages; otherwise, as in *Figure 4B*.

distribution of poly(A)-tail lengths remained unimodal as overall lengths increased, which indicated that the focus on the mean tail-length change in our analyses did not miss specific regulation of a subset of mRNAs for a gene (*Figure 4B*). Also helpful for considering the effects of increasing tail lengths during oocyte maturation is the steep relationship between poly(A)-tail length and TE during this developmental period (*Figure 2*). The steepness of this relationship in stage 11, 12, and 13 oocytes implied a 6- to 7-fold increase in the median TE for mRNAs with a poly(A)-tail length between 70–80 nt relative to those with a poly(A)-tail length between 30–40 nt. In stage 14 oocytes, the poor correlation between tail length and TE resulted in only a 1.8-fold increase in median TE when comparing mRNAs with such tail lengths. Thus during oocyte maturation, the modest increases observed in mean poly(A)-tail lengths presumably explain part, but not all, of the robust translational activation of *cyclin B, cyclin B3, fizzy*, and mRNAs from many other genes. Moreover, a broad range of TE values was observed for mRNAs with the same mean poly(A)-tail length, presumably as a consequence of additional regulatory processes that are independent of poly(A)-tail length. To the extent that these additional processes act concordantly with tail-length–dependent processes, they would increase the apparent effect of tail lengthening.

## Translational regulation during the oocyte-to-embryo transition

Although egg activation occurs rapidly (within ~20 min) (*Horner and Wolfner, 2008*), this developmental transition induced TE changes for mRNAs from many genes, some of which exceeded 30 fold (*Figure 2*, lower right) (*Kronja et al., 2014b*). As in the preceding period of oocyte maturation,

tail-length changes corresponded to TE changes (*Figure 4A and C*). Indeed, during egg activation the correlation between tail-length change and TE change exceeded that observed for any other transition (*Figure 1B* and *Figure 2*, lower panels; *Figure 4A*). Thus during egg activation, changes in poly(A)-tail length can account for a large proportion of the translational changes.

mRNAs for the embryonic patterning genes *bicoid, torso*, and *Toll* are translationally upregulated during egg activation in a manner reported to depend on changes in poly(A)-tail length (*Salles et al., 1994*). In our analyses, translation of *bicoid* mRNA was undetectable or detected with very few reads in oocytes from stages 11 through 14, but it became much more highly translated following egg activation, increasing by 36 fold in the activated egg relative to the stage 14 oocyte (median TE change, 1.2 fold). mRNAs for *torso* and *Toll* were somewhat better translated in stage 14 oocytes, but nonetheless their TEs increased 13 and 46 fold, respectively (*Figure 4C*). During this time, the *Toll* poly(A)-tail length increased 3.3 fold (median tail length change, 1.1 fold), which corresponded well to the trend observed globally for tail-length and TE changes (*Figure 4C*), suggesting that its increased tail-length might fully account for its increased TE. However, concordantly acting tail-length–independent processes that increase the apparent effect of tail-length changes might also be influencing *Toll* translation. Indeed, our subsequent experiments with *wispy*-mutant animals showed that in the absence of cytoplasmic polyadenylation, *Toll* underwent substantial translational upregulation during egg activation and remained one of the most efficiently translated mRNAs in the *wispy*-mutant laid egg, despite having a mean tail length of only 14 nt. The poly(A)-tail lengths of *bicoid* and *torso* increased by only 1.7 and 1.3 fold, respectively, putting them outside the global trend (*Figure 4C*), and the unimodal tail-length distributions showed again that our focus on the mean change did not miss a unique behavior of a subpopulation of transcripts (*Figure 4D*). Thus, the translational regulation of these mRNAs seems to also involve tail-length–independent mechanisms.

The strong translational upregulation of *bicoid* mRNA, from essentially untranslated in oocytes from stages 11 through 14 to relatively well translated in the activated egg, implied the influence of either a translational repressor acting on *bicoid* mRNA in the oocyte that was relieved at egg activation, or a translational activator acting on it during egg activation. Because *bicoid* mRNA had an average tail length of 42 nt in stage 14 oocytes, which was similar to that of many well translated mRNAs at this stage, we conclude that there was likely a translational repressor acting on it in oocytes to prevent its translation prior to egg activation. The same is presumably true for other mRNAs that had little to no detectable translation in oocytes but then became efficiently translated following egg activation. For mRNAs such as *torso*, which had somewhat greater translation in stage 14 oocytes but subsequently experienced a TE increase that was out of proportion to its tail-length increase, either possibility might account for its translational upregulation at egg activation. Likewise, the 15-fold and 4-fold translational activation observed between stage 14 oocytes and 0–1 hr embryos for *zelda* and *stat92E*, respectively (*Figure 1C*), were accompanied by only 1.2-fold changes in poly(A)-tail length for both mRNAs (median TE change, 1.2 fold; median tail length change, 1.1 fold), again implying that a component of their activation may have occurred through a tail-length independent mechanism (note that the poly(A)-tail length of *zelda* was quantified by only 31 and 72 tags in stage 14 oocytes and 0–1 hr embryos, respectively).

## Widespread impact of cytoplasmic polyadenylation on poly(A)-tail lengths but not TE

During oocyte maturation and egg activation, changes in poly(A)-tail length occur through the opposing activities of deadenylases and cytoplasmic poly(A)-polymerases, which shorten and lengthen poly(A) tails, respectively (*Weill et al., 2012*). Wispy is the cytoplasmic poly(A)-polymerase responsible for poly(A)-tail lengthening at the end of oocyte maturation and continuing through egg activation (*Benoit et al., 2008*; *Cui et al., 2008*). Indeed, within this developmental window, most maternal transcripts bind to oligo(dT) with less affinity in *wispy* mutants, indicating that many poly(A) tails shorten in the absence of Wispy (*Cui et al., 2013*), but the degree to which tail-lengths shorten and the impact on TE had not been previously measured.

Our PAL-seq measurements of poly(A)-tail lengths in *wispy*-mutant stage 13 oocytes and laid eggs confirmed the widespread shortening of poly(A) tails in the absence of Wispy (*Figure 5A*). Of the mRNAs from 3571 genes measured in stage 13 oocytes, 1801 had tails that were at least 50% shorter in *wispy*-mutant oocytes relative to wild-type, and for all but a few mRNAs the poly(A) tail

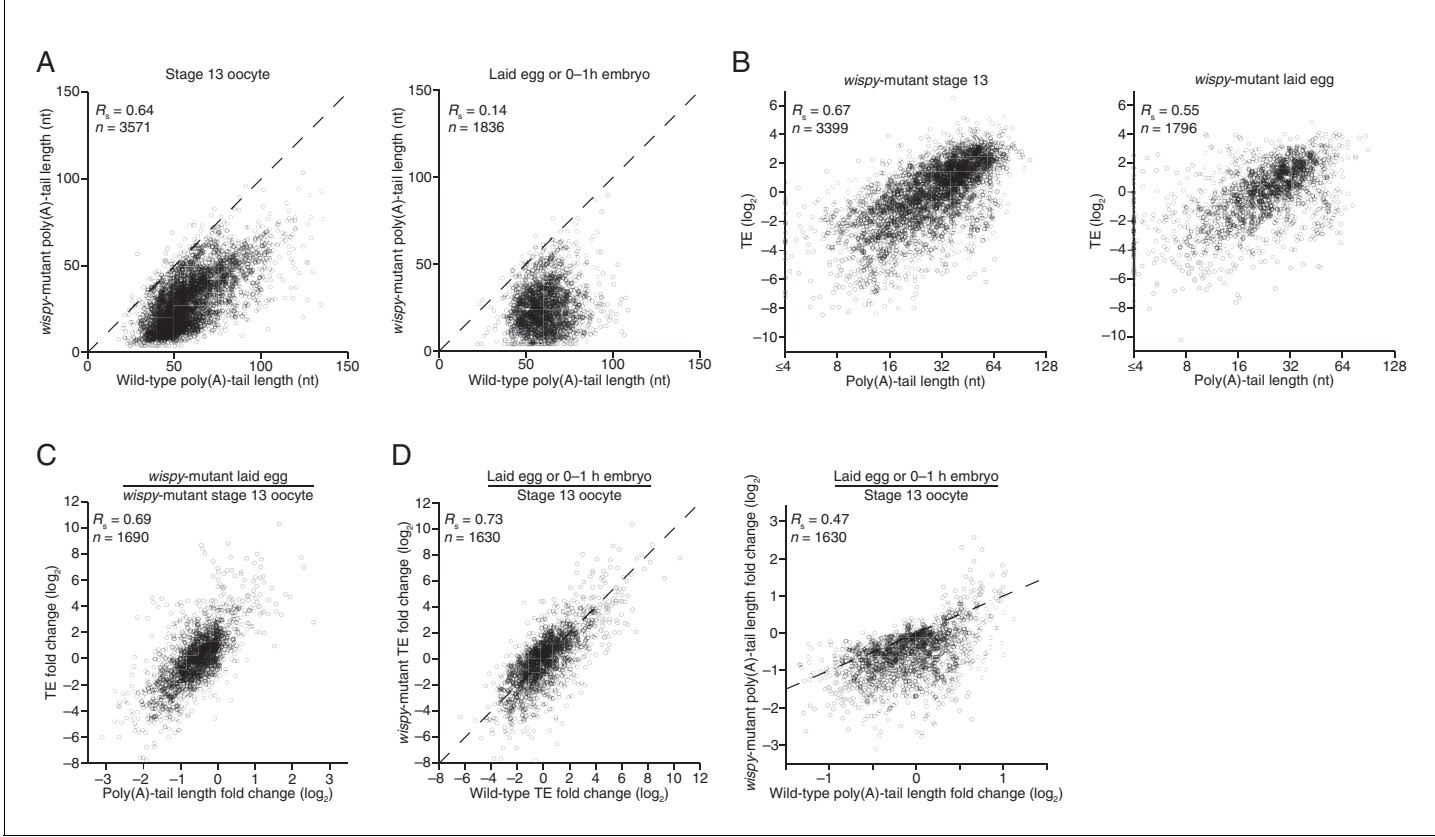

**Figure 5.** Widespread impact of Wispy on poly(A)-tail length but not TE. (**A**) Comparison of mean poly(A)-tail lengths in wild-type and *wispy*-mutant stage 13 oocytes (*left*) and in wild-type cleavage-stage embryos and *wispy*-mutant laid eggs (*right*). Plotted are mean poly(A)-tail lengths of mRNAs with ≥100 poly(A) tags in both the wild-type and *wispy*-mutant sample at the indicated developmental stage. mRNAs that had mean tail-length values ≤4 nt are reported as 4 nt. The dashed line is for y = x. (**B**) Relationship between mean poly(A)-tail length and TE at the indicated developmental stage in *wispy*-mutant oocytes (*left*) and laid eggs (*right*). The TEs were calculated by dividing *wispy*-mutant stage 13 oocyte or laid egg RPF data by wild-type stage 13 oocyte or 0–1 hr embryo RNA-seq data, respectively. TE values (log$_2$) were median centered (median values in *wispy*-mutant stage 13 oocytes and laid eggs, –1.0551 and 0.3302, respectively). mRNAs that had mean tail-length values ≤4 nt are reported as ≤4 nt; otherwise, as in **Figure 1B**. (**C**) Relationship between tail-length and TE changes observed between *wispy*-mutant laid eggs and stage 13 oocytes. TE fold-change values (log$_2$) were median centered (median value, 0.8132); otherwise, as in **Figure 1B**. The mRNAs that seemed to have increased poly(A)-tail lengths over this time tended to have very short poly(A)-tails in stage 13 oocytes, suggesting that their positive fold-change values reflected difficulties in accurately measuring poly(A)-tails <8 nt using PAL-seq rather than genuine increases in poly(A)-tail length. (**D**) Comparison of TE changes for wild-type and *wispy*-mutant samples during the OET (*left*) and comparison of tail-length changes for wild-type and *wispy*-mutant samples during the OET (*right*). TE fold-change values (log$_2$) were median centered (median values for the wild-type and wispy-mutant samples, 0.7207 and 0.8367, respectively). Dashed line is for y = x. Otherwise, this panel is as in **Figure 1B**.

The following figure supplement is available for figure 5:

**Figure supplement 1.** Impact of Wispy on the poly(A)-tail lengths, mRNA recovery, and RPFs.

was at least somewhat shorter in the absence of Wispy (**Figure 5A**). Although *wispy*-mutant eggs do not complete meiosis and thus do not begin nuclear divisions, most of the molecular changes characteristic of the OET are independent of these divisions, motivating a comparison of *wispy*-mutant laid eggs with wild-type early embryos. In *wispy*-mutant laid eggs, 1219 of the 1836 measured mRNAs had poly(A) tails at least 50% shorter than those in the analogous wild-type sample (0–1 hr embryos), and again for all but a few mRNAs the poly(A) tail was shorter in the absence of Wispy (**Figure 5A**). Examination of the mRNAs with tails that were least affected by the absence of Wispy revealed a remarkable enrichment for mRNAs of ribosomal proteins, both in oocytes and laid eggs (**Figure 5—figure supplement 1A**; p<10$^{-41}$ and p<10$^{-44}$, respectively, one-tailed Wilcoxon rank sum test).

Strikingly, the global changes in poly(A)-tail length that occurred in the absence of Wispy did not affect the coupling between poly(A)-tail length and TE. In *wispy*-mutant stage 13 oocytes and laid eggs, poly(A)-tail length and TE were correlated to a similar extent as in wild-type oocytes and embryos (*Figure 1B*, *2*, and *Figure 5—figure supplement 1B*). These correlations were observed even though the RNA-seq measurements used to calculate TEs were made using poly(A)-selected RNA, which in the *wispy*-mutant samples likely resulted in inflated TE estimates for mRNAs with very short poly(A)-tails, as these would be poorly recovered during the poly(A) selection. Indeed, RNA-seq measurements correlated with poly(A)-tail lengths in the *wispy*-mutant samples but not the wild-type samples (*Figure 5—figure supplement 1C*), indicating a poly(A)-selection bias in the *wispy*-mutant RNA-seq data but not in the wild-type data. Because neither developmental stage being considered had appreciable transcription or RNA degradation, and because of the poly(A)-selection bias in the *wispy*-mutant RNA-seq data, we reasoned that the RNA-seq measurements from corresponding wild-type samples should more accurately reflect mRNA abundances in the *wispy*-mutant samples. Indeed, results obtained from determining TEs for the *wispy*-mutant samples using the *wispy*-mutant ribosome profiling and the wild-type RNA-seq (*Figure 5B*) were similar to those obtained when only *wispy*-mutant measurements were used (*Figure 5—figure supplement 1B*), except they had somewhat stronger correlations between poly(A)-tail length and TE, which supported the idea that these modified TEs more accurately reflected the true TEs at these stages.

Importantly, the strong correlation between changes in poly(A)-tail length and changes in TE that occurred at egg activation in wild-type flies was retained in *wispy*-mutant flies (*Figures 1B* and *5C*, respectively), indicating that changes in the length of the poly(A) tail—not the act of cytoplasmic polyadenylation per se—is what influences TE. Moreover, the changes in TE that occurred during egg activation in *wispy*-mutant samples strongly correlated with those in the wild-type samples, as did the relative changes in poly(A)-tail length (*Figure 5D*). Thus, although Wispy lengthens the poly(A)-tails of most mRNAs during egg activation, our results indicated that this lengthening was largely dispensable for reshaping TEs during egg activation, as the rank order of tail-length and TE changes at this time were largely preserved in the *wispy*-mutant samples.

The observation that cytoplasmic polyadenylation was not needed to generate the rank order of tail-length and TE changes for most mRNAs warranted further investigation, particularly when considering both the critical influence of cytoplasmic polyadenylation for generating these differences during vertebrate development and the importance of Wispy for relieving meiotic arrest during the OET. Perhaps some mRNAs most affected by the loss of Wispy dropped out of our tail-length analysis because they had very short poly(A)-tails in the *wispy* mutant and thus were poorly recovered in our PAL-seq libraries. Although the tail lengths of such mRNAs might have been difficult to measure by PAL-seq, the RPF measurements of mRNAs for all genes can be analyzed without consideration of the corresponding tail-length or mRNA abundance measurements, either of which might selectively lose the set of mRNAs most dependent on Wispy for their regulation. The RPFs in wild-type and *wispy*-mutant stage 13 oocytes were well correlated, and the same was true for 0–1 hr wild-type embryos compared with *wispy*-mutant laid eggs ($R_s$ = 0.76 for both, *Figure 5—figure supplement 1D*). Although globally the RPFs were largely preserved in the *wispy*-mutant samples, some mRNAs had many RPFs in the wild-type samples but none in the *wispy*-mutant samples (*Figure 5—figure supplement 1D*). Included among these was *cortex*, a known target of Wispy, the upregulation of which is essential for the OET (*Chu et al., 2001*; *Benoit et al., 2008*).

We conclude that during egg activation in Drosophila, cytoplasmic polyadenylation extends the tails of nearly all mRNAs but does not play the major role in generating the relative differences in tail lengths and TEs observed between mRNAs, which implies that poly(A)-tail shortening instead plays the major role in generating these differences. Nonetheless, the mRNAs responsible for the *wispy*-mutant phenotype might be among those that are most strongly dependent on Wispy for their translational state.

## PNG kinase complex up- and downregulates translation through changes in poly(A)-tail length

The PNG kinase, which is required for nuclear divisions in the early embryo, influences the translational up- or downregulation of hundreds of different mRNAs at egg activation (*Figure 6A*) (*Kronja et al., 2014b*), presumably through its regulation of translational regulators, such as SMG (*Tadros et al., 2007*). To characterize the extent to which these downstream regulators might use

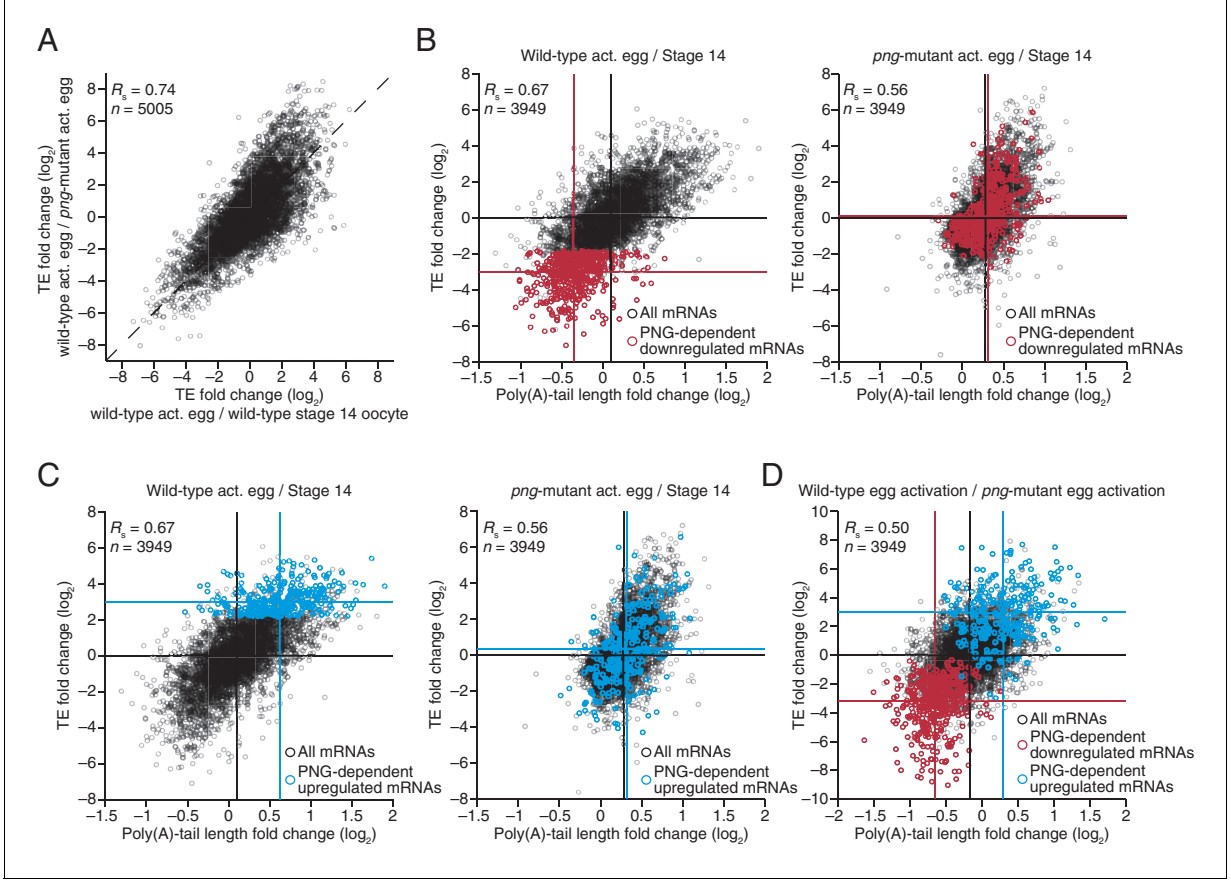

**Figure 6.** Widespread translational regulation by PAN GU primarily attributable to changes in poly(A)-tail length. (A) Relationship between the TE changes in wild-type activated eggs relative to wild-type stage 14 oocytes and those in wild-type activated eggs relative to *png*-mutant activated eggs. TE fold-change values (log₂) were median centered (median values for wild-type activated egg relative to stage 14 oocyte and for wild-type activated egg relative to *png*-mutant activated egg, 0.3065 and 0.0881, respectively). Results are plotted for all mRNAs that had ≥10.0 RPM in the RNA-seq data of all samples, and ≥10.0 RPM in the ribosome profiling data of at least one of the two samples being compared and >0.0 RPM in the other. mRNAs ≥4-fold up- or downregulated in both comparisons were defined as the PNG-dependent up- or downregulated mRNAs, respectively. The TE data for wild-type stage 14 oocytes and activated eggs and *png*-mutant activated eggs were from *Kronja et al. (2014b)*. Dashed line is for y = x. (B) Relationship between mean tail-length changes and TE changes after egg activation for wild-type activated eggs (*left*; modified from *Figure 2* to include only the mRNAs that also passed the cutoffs for the *png*-mutant comparison) and *png*-mutant activated eggs (*right*). The mRNAs with PNG-dependent downregulation are highlighted in red (*Supplementary file 3*), analyzing and highlighting the same mRNAs in both plots. TE fold-change values (log₂) were median centered (median for wild-type and *png*-mutant samples, 0.1366 and 0.6448, respectively. Black lines indicate median values for all mRNAs, and red lines indicate median values for the PNG-dependent downregulated mRNAs. The TE data for wild-type stage 14 oocytes and activated eggs and *png*-mutant activated eggs are from *Kronja et al. (2014b)*. Otherwise, this panel is as in *Figure 1B*. (C) The plots of panel B, highlighting the PNG-dependent upregulated mRNAs in blue (*Supplementary file 3*). (D) Relationship between tail-length and TE changes during egg activation in wild-type relative to the *png*-mutant samples, analyzing and highlighting the same mRNAs as in panels B and C. TE fold-change values (log₂) were median centered (median value, –0.727). Otherwise, this panel is as in *Figure 6B*.

The following figure supplement is available for figure 6:

**Figure supplement 1.** Perturbation of tail-length and TE during egg activation in *png*-mutant samples.

tail-length changes to influence TE, we profiled poly(A)-tail lengths and TEs in *png*-mutant stage 14 oocytes and activated eggs. Poly(A)-tail lengths and TEs were highly correlated between *png*-mutant and wild-type stage 14 oocytes, whereas the correlation was substantially lower between *png*-mutant and wild-type activated eggs (*Figure 6—figure supplement 1A*). Additionally, the poly(A)-tail lengths and TEs in *png*-mutant activated eggs were quite similar to those in *png*-mutant stage 14 oocytes, whereas this was not the case for the same comparisons among wild-type samples

(*Figure 6—figure supplement 1B*). The requirement of PNG for achieving the gene-expression profile of the wild-type activated egg but not the stage 14 oocyte was consistent with PNG kinase becoming active during egg activation to serve as a master regulator of the OET (*Shamanski and Orr-Weaver, 1991*; *Kronja et al., 2014b*).

The set of mRNAs dependent on PNG for their translational downregulation was predominantly those that underwent tail-length shortening during egg activation in wild-type samples (median TE $\log_2$ fold change of –3.03 after median centering and median tail-length $\log_2$ fold change of –0.35) (*Figure 6B*). In *png*-mutant embryos, these mRNAs failed to be deadenylated, as expected if tail-length shortening mediated their translational repression (median TE $\log_2$ fold change of 0.10 after median centering and median tail-length $\log_2$ fold change of 0.32, which closely resembled the median value for all mRNAs) (*Figure 6B*). Correspondingly, the set of mRNAs dependent on PNG for their translational upregulation was predominantly those that underwent tail-length extension during wild-type egg activation (median TE $\log_2$ fold change of 2.98 after median centering and median tail-length $\log_2$ fold change of 0.62), and these mRNAs failed to undergo tail lengthening in *png* mutants, again as expected if tail-lengthening mediated their translational activation (median TE $\log_2$ fold change of 0.36 after median centering and median tail-length $\log_2$ fold change of 0.32, which closely resembled the median value for all mRNAs) (*Figure 6C*). mRNAs that were translationally up- or downregulated by PNG also typically increased or decreased in poly(A)-tail length, respectively (*Figure 6D*), and the tail-length changes for either the up- or downregulated sets of mRNAs were largely dependent on PNG (*Figure 6B and C*), which showed that PNG plays a central role in regulating polyadenylation and deadenylation for the hundreds of mRNAs present at egg activation.

## SMG-mediated repression is primarily attributable to tail shortening and explains some but not all of the PNG-mediated translational repression

Because much of the PNG-mediated translational repression is attributed to its derepression of *smg* translation (*Tadros et al., 2007*), we examined the impact of SMG on the set of mRNAs that depended on PNG for tail-length shortening and translational repression. SMG is reported to downregulate translation of its target mRNAs through either direct recruitment of the deadenylation machinery or recruitment of CUP, which directly represses translation initiation and can also promote deadenylation (*Nelson et al., 2004*; *Semotok et al., 2005*; *Igreja and Izaurralde, 2011*). Our results showed that in the absence of SMG, the poly(A)-tail lengths for the set of PNG-dependent downregulated mRNAs were less shortened during egg activation (median tail-length $\log_2$ fold changes of –0.22 and –0.04 for the wild-type and *smg* mutant, respectively, relative to the median of all mRNAs, $p<10^{-27}$, one-tailed Wilcoxon rank sum test), and this coincided with their partial translational derepression (median TE $\log_2$ fold change of –2.2 and –1.2, respectively, after median centering, $p<10^{-32}$, one-tailed Wilcoxon rank sum test) (*Figure 7A*). In contrast, no significant difference was observed for either poly(A)-tail length or TE changes for the PNG-dependent upregulated mRNAs (median tail-length $\log_2$ fold changes of 0.27 for both the wild-type and *smg* mutant relative to the median of all mRNAs, and median TE $\log_2$ fold changes of 2.8 and 2.7, respectively, after median centering for the wild-type and *smg* mutant, $p=0.89$ and 0.37 for tail-length or TE changes, respectively, two-tailed Wilcoxon rank sum test), consistent with SMG functioning only as a repressor (*Figure 7B*). In general, mRNAs that were translationally repressed by SMG also decreased in poly-(A)-tail length, which indicated that recruitment of deadenylases was the dominant mechanism of SMG-dependent translational repression occurring during egg activation and early embryogenesis (*Figure 7C*). Additionally, in the absence of SMG, a set of mRNAs reported to be bound by SMG in the early embryo (*Chen et al., 2014*) had longer poly(A) tails and were better translated following the OET (median tail-length $\log_2$ fold changes of –0.21 and –0.04 for the wild-type and *smg* mutant, respectively, relative to the median of all mRNAs, and median TE $\log_2$ fold changes of –1.0 and –0.15, respectively, after median centering, $p<10^{-16}$ and $10^{-11}$ for tail length or TE changes, respectively, one-tailed Wilcoxon rank sum test) (*Figure 7—figure supplement 1*).

Taken together, our results indicate that SMG mediates much of the PNG-dependent translational repression occurring during egg activation and early embryogenesis, with most of this repression attributable to poly(A)-tail shortening. Nevertheless, the residual repression observed in the

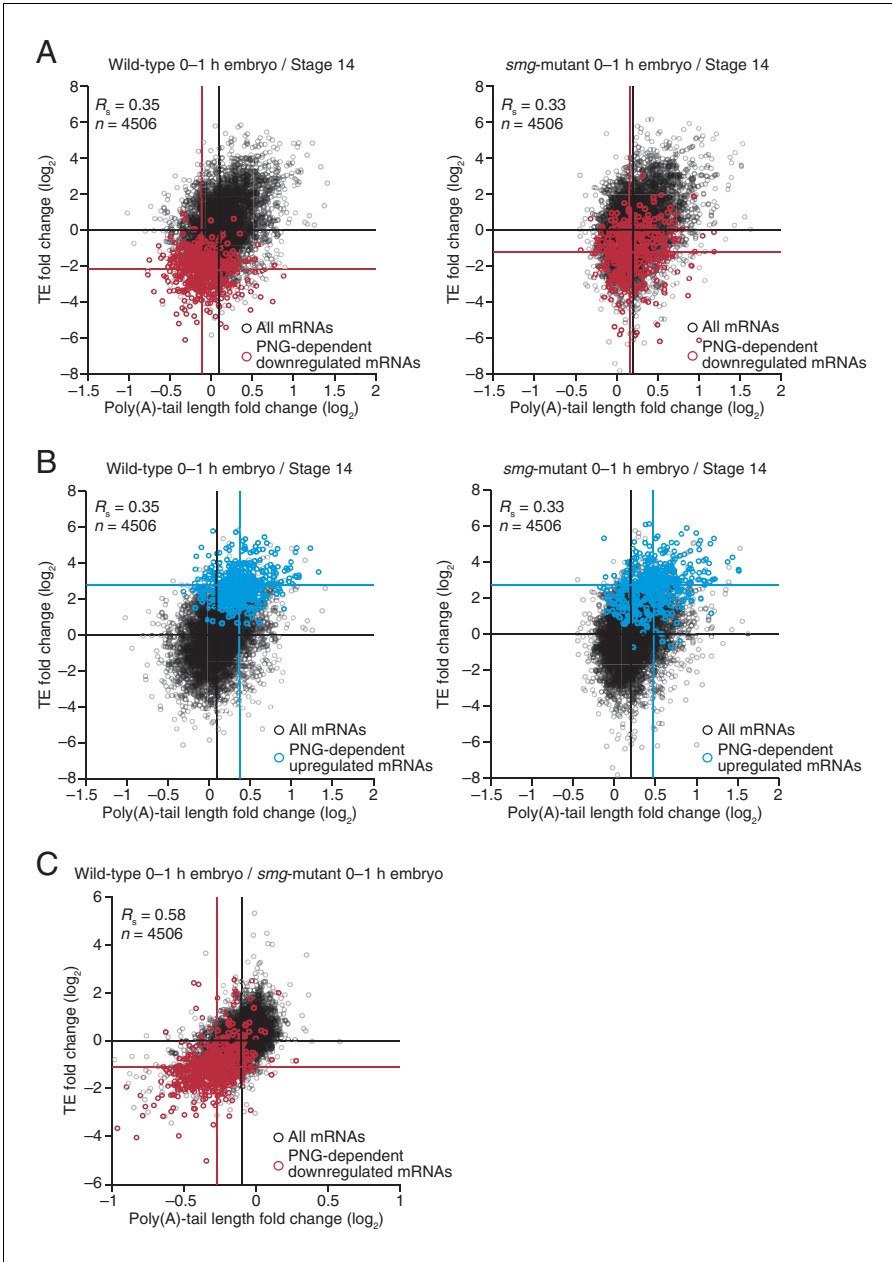

**Figure 7.** Translational regulation by Smaug primarily explained by changes in poly(A)-tail length. (**A**) Relationship between mean tail-length changes and TE changes during the OET for wild-type 0–1 hr embryos (*left*) and *smg*-mutant 0–1 hr embryos (*right*), comparing wild-type or *smg*-mutant 0–1 hr embryos to wild-type stage 14 oocytes. The wild-type plot is redrawn from *Figure 1B*, but includes only the mRNAs that also passed the cutoffs for the smg-mutant comparison. TE fold-change values ($log_2$) were median centered (median for the wild-type and *smg*-mutant samples, 0.175 and –0.0221, respectively). The PNG-dependent downregulated mRNAs are in red (*Supplementary file 3*), analyzing and highlighting the same mRNAs in both plots. Otherwise, this panel is as in *Figure 6B*. (**B**) The plots of panel **A**, highlighting the PNG-dependent upregulated mRNAs in blue (*Supplementary file 3*). (**C**) Relationship between tail-length and TE changes for wild-type 0–1 hr embryos relative to *smg*-mutant 0–1 hr embryos, analyzing and highlighting the same mRNAs as in panel **A**. TE fold-change values ($log_2$) were median centered (median value, 0.2634). Otherwise, this panel is as in *Figure 6D*.

The following figure supplements are available for figure 7:

**Figure supplement 1.** Smaug-dependent translational repression of Smaug binding targets, primarily explained by changes in poly(A)-tail length.

*Figure 7 continued on next page*

*Figure 7 continued*

**Figure supplement 2.** Relationship between poly(A)-tail length changes and TE changes throughout development, plotting absolute rather than relative tail-length changes.

absence of SMG (*Figure 7A*) indicates that one or more additional PNG-activated translational repressors also operate at this time.

## Discussion

Our results from Drosophila oocytes and embryos increase the number of cellular environments known to possess global coupling between poly(A)-tail length and TE, which now include several stages of Drosophila oocytes as well as Drosophila, zebrafish, and Xenopus embryos prior to the onset of gastrulation. At least as early as the primary meiotic arrest point, changes in poly(A)-tail length largely explain changes in TE, and this regulatory strategy continues through early embryonic development. A switch around the onset of gastrulation uncouples poly(A)-tail length and TE in Drosophila, mirroring the developmental timing of the same uncoupling in zebrafish and Xenopus embryos. The uncoupling of TE from poly(A)-tail length is thus a broadly conserved developmental switch in the nature of translational control, occurring around the onset of gastrulation in both vertebrate and invertebrate embryos.

Changing the length of a poly(A) tail would influence both the number of poly(A)-binding proteins (PABPs) that could bind the tail and the probability that at least one PABP would be bound, with the former being more relevant in conditions of saturating PABP and the later being more relevant in conditions in which tails are competing for PABP. If the number of PABPs that bind the tail were most relevant, then TE changes might have correlated better with absolute tail-length changes than with relative tail-length changes. However, repeating our analyses using absolute tail-length changes rather than tail-length $\log_2$ fold changes yielded the same results for the respective comparisons, with nearly identical $R_s$ values (*Figure 7—figure supplement 2*), suggesting that PABP is limiting in the developmental stages in which tail length and TE are coupled, such that mRNAs are competing for PABP.

We presume that the correlation between poly(A)-tail length and TE that we observed in oocytes and pre-gastrulation embryos reflects a causal relationship, i.e., that changes in poly(A)-tail length at least partially cause the corresponding changes in TE. The other possibilities are that changes in TE cause changes in tail length or that these two measurements merely correlate without having a causal relationship. Because previous single-gene studies clearly demonstrate a causal role for cytoplasmic polyadenylation, and poly(A)-tail length itself, in regulating the translation of mRNAs during these developmental times (*McGrew et al., 1989*; *Salles et al., 1994*; *Barkoff et al., 1998*), we interpret our observation of a global correlation between poly(A)-tail length and TE as reflecting a causal relationship between the two wherein tail-length changes cause TE changes. Given this interpretation, our results indicate that widespread changes in poly(A)-tail length broadly reshape the translational profiles of oocytes and early embryos.

Despite this strong qualitative conclusion, we cannot provide quantitative estimates of the proportion of each TE change that is attributable to each tail-length change, as additional translational regulatory mechanisms presumably act to reinforce or oppose the effects of tail-length–dependent translational regulation. Indeed, we suspect that tail-length–independent processes often act concordantly with tail-length–dependent processes and thereby amplify the apparent effects of tail-length changes. Such amplification would explain why some mRNAs, including *cyclin B, cyclin B3, fizzy, zelda, stat92E, bicoid*, and *torso*, undergo translational activation that seems out of proportion to their tail-length changes.

A primary motivation for using Drosophila in our study was the opportunity to examine the impact of Wispy, PNG, and SMG on poly(A)-tail length and TE. Of particular interest was Wispy, the cytoplasmic poly(A) polymerase that acts during late oogenesis and early embryogenesis, and is essential for the OET (*Benoit et al., 2008*; *Cui et al., 2008*). Analysis of *wispy* mutants revealed key insights into both the mechanism of translational activation and the importance of tail shortening (as

opposed to tail lengthening) in specifying translational control in Drosophila maturing oocytes and early embryos. Although poly(A)-tail lengths were typically shorter in *wispy*-mutant stage 13 oocytes and laid eggs compared to the corresponding wild-type samples, tail length and TE were coupled in both of these *wispy*-mutant contexts, and the relative changes in poly(A)-tail lengths during the OET in wild-type samples were largely preserved in the *wispy*-mutant samples, as were the changes in TE. Thus, the act of cytoplasmic polyadenylation is not necessary for the coupling of poly(A)-tail length to translation. Moreover, cytoplasmic polyadenylation is largely dispensable for specifying which mRNAs are to be translationally activated or repressed, implying that in Drosophila, selective deadenylation is more important for imparting this specificity. This greater importance for deadenylation does not preclude a role for selective cytoplasmic polyadenylation in translational control in Drosophila—differences were observed when comparing TE changes in wild-type and *wispy*-mutant samples (*Figure 5D*), and when comparing RPF measurements between wild-type and *wispy*-mutant samples (*Figure 5—figure supplement 1D*), including for *cortex*, a critical OET regulator. However, the greater importance of selective deadenylation for specifying translational control in Drosophila differs from the prevailing paradigm of vertebrates, which centers on selective cytoplasmic polyadenylation of mRNAs with cytoplasmic polyadenylation elements (*Piqué et al., 2008*; *Weill et al., 2012*). This suggests either a fundamental difference between invertebrates and vertebrates or that an underappreciated component of translational control in vertebrates occurs through more global cytoplasmic polyadenylation with selective deadenylation.

Despite the surprisingly modest perturbation of translational regulation that we observed in *wispy*-mutant laid eggs during egg activation, these laid eggs are blocked at the OET because of defective progression through meiosis and failure of pronuclei to fuse (*Benoit et al., 2008*; *Cui et al., 2008*). Why does this developmental arrest occur if translational regulation is largely preserved during egg activation in *wispy* mutants? One possibility is that the absolute level of translation in *wispy*-mutant eggs is reduced such that they make an inadequate amount of one or more essential protein. The other possibility rests on the observation that although TE changes and RPFs were remarkably concordant in the *wispy*-mutant and wild-type samples, they were not identical, and thus an aberrant change in expression for one or more key factor, such as *cortex*, might be sufficient to trigger the developmental arrest of *wispy*-mutant laid eggs.

Activity of the PNG kinase during egg activation affects most mRNAs that undergo translational activity changes at this developmental transition, both those that are up- and downregulated (*Kronja et al., 2014b*). Although the direct targets of PNG kinase remain to be identified, PNG is required to relieve the repressive effects of PUM (*Vardy and Orr-Weaver, 2007a*) and thereby derepresses translation of *smg* and other mRNAs (*Tadros et al., 2007*). In wild-type activated eggs the mRNAs dependent on PNG for translational up- or downregulation exhibited corresponding increases or decreases in poly(A)-tail length, and these tail-length changes did not occur in *png* mutants. Thus, for both translationally activated and translationally repressed mRNAs, the downstream effectors of PNG appear to regulate translation primarily through changes in poly(A)-tail length.

Translational repressors in addition to SMG have been proposed to contribute to PNG-dependent regulation (*Tadros et al., 2007*). Our results are consistent with the idea that PNG activates at least one additional translational repressor, as the set of mRNAs dependent on PNG for their translational downregulation were still somewhat translationally repressed and deadenylated relative to all other mRNAs in *smg*-mutant embryos, although less than in wild-type embryos. Additionally, although SMG is reported to act partly through a mechanism that does not involve deadenylation (*Nelson et al., 2004*; *Semotok et al., 2005*), essentially all the derepression observed in the *smg*-mutant embryos was accompanied by tail-length increases. Perhaps at other times SMG causes direct translational repression, but in 0–1 hr embryos it seems to act predominantly through deadenylation.

Within the overall framework in which tail-length changes cause the up- or downregulation of mRNAs for a vast repertoire of different genes, we discovered unanticipated complexities and exceptions. One unanticipated complexity was the evidence for tail-length–independent processes that appear to amplify the tail-length–dependent effects, as already discussed. Additional unanticipated complexity was observed during oocyte maturation, in which surprisingly diverse patterns of regulatory dynamics produce the ultimate changes in TE. Single-gene studies in Xenopus, which show that *c-mos* mRNA becomes polyadenylated and translated prior to the mRNAs for several

*cyclin B* genes (*Richter, 2007*; *Piqué et al., 2008*), hint that the complex landscape of translational control observed in Drosophila extends to vertebrate oocyte maturation. The diverse temporal behaviors of translational regulation during oocyte maturation might reflect the evolving demands on the maturing oocyte, or perhaps simply differences in the timing of expression of RNA-binding proteins that interact with these different sets of mRNAs. Regardless, the observation of different temporal patterns of regulation indicates that rather than being a sudden switch, oocyte maturation in Drosophila is a progressive process, whereby mRNAs are translationally activated or repressed at different times throughout this developmental transition.

Another surprise was the identification of a set of mRNAs for which tail-length changes did not correspond to TE changes during oocyte maturation. This exception to the general pattern occurred for the mRNAs that had undergone deadenylation between oocyte stages 13 and 14 yet continued to be translated with relatively high efficiency (*Figure 2—figure supplement 2*). To escape the general regulatory environment of the cytoplasm, these mRNAs might exploit unique interactions with the translational machinery analogous to the translational activation of histone mRNAs through stem-loop binding protein (*Cakmakci et al., 2008*), or they might localize to a subcellular compartment that lacks coupling. The deadenylation of these mRNAs during oogenesis might prime them for rapid degradation during embryogenesis. Most mRNAs from genes encoding proteasome subunits fell into this class. Previous comparisons of the proteome and translation changes suggest that extensive proteolysis occurs at egg activation (*Kronja et al., 2014b*). Upregulation of the proteasome in late oogenesis might necessitate a mechanism to downregulate it rapidly during embryogenesis, perhaps facilitated by pre-deadenylating the corresponding mRNAs before egg activation.

The molecular mechanism that couples TE with poly(A)-tail length is unknown, as is the mechanism of the uncoupling that occurs at gastrulation. Nevertheless, our results, which show that widespread changes in poly(A)-tail length broadly reshape translational activity during oocyte and embryo development, demonstrate the importance of this coupling mechanism and provide a resource that documents the thousands of affected genes with information on the timing, magnitude, and inferred consequences of the tail-length changes for mRNAs from each of these genes. Moreover, the identification of unanticipated complexities and exceptions reveals the rich posttranscriptional regulatory landscape of oocytes and early embryos and points to additional mechanisms that operate for some mRNAs in specific settings, which can now be targeted for investigation.

## Materials and methods

### Drosophila stocks

All flies were kept at 18, 22 or 25°C according to standard procedure (*Greenspan, 1997*). *Oregon R* (*OrR*) flies were used as a wild-type control. The null *png$^{1058}$* allele was previously described (*Shamanski and Orr-Weaver, 1991*; *Fenger et al., 2000*). The *wispy* hemizygote mutants were derived from a cross between *wisp$^{41}$/FM6* (*Cui et al., 2008*) (kindly provided by Mariana Wolfner, Cornell) and *Df(1)RA47/FM7c* flies (BL961, obtained from the Bloomington Drosophila Stock Center). The *smaug* hemizygote mutants resulted from a cross between *smg$^1$/TM3, Sb[1]* (BL5930, obtained from the Bloomington Drosophila Stock Center) and *Df(3L)Scf-R6, Diap1[1] st[1] cu[1] sr[1] e[s] ca[1]/ TM3, Sb[1]* flies (BL4500, obtained from the Bloomington Drosophila Stock Center). Unfertilized eggs were collected from crosses to *twine$^{HB5}$*mutant males, which fail to make sperm (*Courtot et al., 1992*).

### Oocyte and embryo collection

Egg chambers were hand-dissected in Grace's Unsupplemented Insect Media (Gibco) from three-day old flies that had been fattened for two days with wet yeast at 22°C. Approximately 300 egg chambers were collected per oogenesis stage. Oocyte stages 11 through 14 were distinguished using morphological criteria (*Spradling, 1993*). Activated eggs were collected as previously described (*Kronja et al., 2014b*). Samples were transferred to RPF lysis buffer (10 mM Tris-HCl, pH 7.4, 5 mM MgCl$_2$, 100 mM KCl, 2 mM DTT, 100 μg ml$^{-1}$ cycloheximide, 1% Triton X-100, 500 U ml$^{-1}$ RNasin Plus, and protease inhibitor (1x complete, EDTA-free, Roche); wild-type stage 14 oocyte and activated egg samples were transferred to RPF lysis buffer lacking cycloheximide), homogenized and then centrifuged at 10000 rpm for 5 min.The supernatants were collected and flash frozen in liquid

nitrogen to be stored at –80°C. Ultimately, the yield was approximately 40–50 µg of total RNA in 70–120 µl lysate.

Embryos were collected at 25°C by discarding two initial one-hour collections to avoid collecting embryos held within females for prolonged time. Only the subsequent 0–1 hr collection was then processed for sequencing analysis. The same embryo collection strategy was applied to *smg* or *wispy* mutants, though for the *wispy* mutant the collection was extended to 0–2 hr as one hour was too short to gather a sufficient number of laid eggs for sequencing. To obtain *OrR* 2–3 h, 3–4 hr and 5–6 hr embryo samples, 0–1 hr embryos (collected after two previous one-hour collections were discarded) were incubated in a "humid chamber" at 25°C for additional 2, 3 or 5 hr, respectively.

## Embryo staging

Drosophila females can hold developing embryos or lay unfertilized eggs, and so to ensure that the vast majority of the embryos processed for downstream analyses were in the expected developmental stage, a fraction of each embryo collection (approximately 10–20%) was stained with DAPI. This approach was also used to confirm that the embryos laid by mutant mothers (*png, smg,* or *wispy*) were developmentally arrested as previously described. Briefly, collected embryos were dechorionated in 50% bleach for 3 min at room temperature, extensively washed with water, dried and then transferred into a scintillation vial that contained methanol. To remove the vitelline membrane from the embryos, an equal volume of heptane was added and the vial was vigorously shaken by hand for 2 min. Next, embryos were fixed overnight in methanol at 4°C. Embryos were then sequentially rehydrated in a mix of methanol:PBS (9:1, 3:1, 1:1, 1:3) for at least 5 min per step. After a wash in PBS, embryos were incubated for 15 min in a 1 µg ml$^{-1}$ solution of DAPI in PBS, followed by 30 min wash in PBS containing 0.1% Triton X-100. Finally, after removing all the washing solution, samples were mounted in Vectashield.

## Transcript models

Reference transcript annotations were downloaded from the UCSC Genome browser (dm6 in refFlat format), and for each gene the longest transcript was chosen as the initial representative transcript model. Non-coding genes and overlapping genes on the same strand were removed from this analysis. PAL-seq tags from all wild-type samples were combined, filtered to remove tags with a poly(A) tail <20 nt to ensure they represent genuine cleavage and polyadenylation sites, and then aligned with STAR (*Dobin et al., 2013*). 3' UTRs were annotated by identifying positions where ≥2 PAL-seq tags aligned at the same position between the stop codon of one gene and the transcription start site of the neighboring gene on the same strand. Isoforms were annotated as described previously (*Jan et al., 2011*), using a 60 nt window and a maximum 3'-UTR length of 4100 nt (so as to not exceed the 99[th] percentile of previously annotated fly 3' UTRs). If a new 3' terminus was annotated that was distal to the initial model, the 3' end of the model was extended to the distal isoform. If not, the initial representative transcript model was used. The resulting set of representative transcript models (referred to as 'mRNAs') used for our analyses is available at http://www.ncbi.nlm.nih.gov/geo under accession number GSE83616. Models for which the selected 3'-end isoform (either the initial refFlat 3' end or an extension based on our re-annotation) overlapped with a snoRNA or snRNA were included in the analysis of sequencing data but excluded from figures.

## Ribosome footprint profiling, RNA-seq, and PAL-seq

Samples were split and prepared for ribosome-footprint profiling and RNA-seq as described (*Kronja et al., 2014b*), as well as for PAL-seq as described (*Subtelny et al., 2014*). Detailed protocols are available at http://bartellab.wi.mit.edu/protocols.html. The wild-type stage 14 oocyte and activated egg samples were prepared using RPF lysis buffer and sucrose gradients that did not contain cycloheximide, all other samples were prepared using RPF lysis buffer and sucrose gradients that contained cycloheximide. Because TE changes between wild-type stage 14 oocytes and activated eggs for samples prepared with and without cycloheximide correlated very well (*Kronja et al., 2014b*), the omission of cycloheximide did not affect our observations pertaining to these stages. RPF and RNA-seq reads were mapped to ORFs of representative transcript models as described, which excluded the first 50 nt of each ORF to eliminate signal from ribosomes that initiated after adding cycloheximide (*Subtelny et al., 2014*). PAL-seq tags were mapped to 3' UTRs of

representative transcript models, and PAL-seq fluorescence intensities were converted to tail lengths as described (*Subtelny et al., 2014*).

## Clustering TE dynamics

TE values (log$_2$) for stage 11, 12, 13, and 14 oocytes, or for stage 14 oocytes, 0–1 hr embryos, and 2–3 hr embryos were normalized by the median TE value for the mRNAs in the corresponding sample from *Figure 2* or *Figure 1B*, respectively, and were then entered into the Short Time-series Expression Miner (STEM) (*Ernst and Bar-Joseph, 2006*), which normalized the data relative to the first sample and then used the STEM clustering method to identify significantly enriched profiles (*p* value < 0.05, following Bonferonni correction). The default settings were used, except a maximum unit change of one in model profiles was allowed between time points, and only mRNAs with $\geq 0.5$ or $\leq -0.5$ log$_2$ fold change between any two samples were placed in a cluster. A heatmap of all expression patterns was generated in Matlab using the normalized data.

## Sequencing data

The RNA-seq and ribosome-footprint profiling data for wild-type stage 14 oocytes and activated eggs, and *png*-mutant activated eggs were previously published (*Kronja et al., 2014b*) and were reanalyzed for this study. The raw data for these previously published datasets are available at the Gene Expression Omnibus (http://www.ncbi.nlm.nih.gov/geo) under accession number GSE52799, and the processed data files used for this study are under accession number GSE83616. The PAL-seq data for those samples had not been published and are available along with all of the other RNA-seq, ribosome profiling, and PAL-seq data and processed data files under accession number GSE83616. All of the processed data are also available in *Supplementary file 2*.

## Statistical analysis

All correlations report the Spearman correlation coefficient, and all statistical tests were two-sided unless noted otherwise. The TE changes observed between biological replicates for wild-type stage 14 oocytes and activated eggs correlated very well ($R_s = 0.94$) (*Kronja et al., 2014b*), even though cycloheximide was omitted from one of the replicates. Likewise, TE of activated egg highly correlated with that of 0–1 hr embryo ($R_s = 0.88$). Although we did not analyze replicates of PAL-seq measurements in the current study, previous analysis of biological replicates demonstrates the reproducibility of mean tail lengths measured using this technique ($R_s = 0.83$). Similar correlations were observed in the current study when comparing stages for which little if any differences in tail lengths were expected, including activated egg and 0–1 hr embryo ($R_s = 0.86$), stage 11 and stage 12 oocyte ($R_s = 0.94$), stage 12 and stage 13 oocyte ($R_s = 0.88$), as well as *png* and wild-type oocytes at stage 14 ($R_s = 0.86$), which was prior to PNG activation (*Supplementary file 1* and *Figure 6—figure supplement 1A*).

## Acknowledgements

We thank Mariana Wolfner for providing stocks, the Whitehead Genome Sequencing Core for sequencing, and Anthony Mahowald for comments on the manuscript. This work was supported by NIH grants GM39341 and GM118098 (TLO-W.) and GM067031 (DPB). AOS was supported by NIH Medical Scientist Training Program fellowship T32GM007753, and IK was supported by the Feodor Lynen Postdoctoral Fellowship from the Alexander von Humboldt Foundation. TLO-W is an American Cancer Society Research Professor, and DPB is an investigator of the Howard Hughes Medical Institute.

## Additional information

### Funding

| Funder | Grant reference number | Author |
| --- | --- | --- |
| National Institutes of Health | Medical Scientist Training Program fellowship T32GM007753 | Alexander Orest Subtelny |

| Alexander von Humboldt-Stif-tung | Feodor Lynen Postdoctoral Fellowship | Iva Kronja |
|---|---|---|
| American Cancer Society | | Terry L Orr-Weaver |
| National Institutes of Health | GM39341 | Terry L Orr-Weaver |
| National Institutes of Health | GM118098 | Terry L Orr-Weaver |
| Howard Hughes Medical Insti-tute | | David P Bartel |
| National Institutes of Health | GM067031 | David P Bartel |

The funders had no role in study design, data collection and interpretation, or the decision to submit the work for publication.

### Author contributions

SWE, AOS, IK, Conception and design, Acquisition of data, Analysis and interpretation of data, Drafting or revising the article; JCK, Analysis and interpretation of data, Drafting or revising the article; TLO-W, DPB, Conception and design, Analysis and interpretation of data, Drafting or revising the article

### Author ORCIDs

Stephen W Eichhorn, iD http://orcid.org/0000-0002-6410-4699

David P Bartel, iD http://orcid.org/0000-0002-3872-2856

## Additional files

### Supplementary files

• Supplementary file 1. Relationships between RNA-seq, ribosome profiling, and PAL-seq measurements for wild-type samples at different developmental stages. The Spearman correlation coefficients for all unique pairwise combinations of stages are shown. For the RNA-seq or ribosome-profiling comparisons, all mRNAs with $\geq 10.0$ RPM in both samples of the respective datasets being compared were included. For the PAL-seq comparisons, all mRNAs with $\geq 100$ poly(A) tags in both samples being compared were included.

• Supplementary file 2. Processed RNA-seq, ribosome-profiling, and PAL-seq data. Each of the 16 spreadsheets of this file reports the data for the indicated sample. Within each sheet, the initial Refseq ID is the dm6 Refseq ID that was selected for each gene on the basis of being the longest annotated isoform of that gene. For many genes, the 3' end of this gene model was extended to include the most distal isoform supported by PAL-seq data. RPKM is reads per kilobase per million mapped reads. Descriptions of the analyses include the measurement cutoffs applied to RNA-seq, ribosome profiling, and PAL-seq data, as applicable. The 8th and 9th columns specify whether a 10 RPM cutoff for RNA-seq or ribosome profiling data was met, and the 10th column specifies whether a 100 tag cutoff for PAL-seq data was met. All transcripts with a 3' UTR that overlapped a snoRNA or snRNA were excluded from analysis, as indicated in the final column.

• Supplementary file 3. Lists of mRNAs highlighted in figures. This file lists mRNAs that are highlighted in *Figures 6* and *7*, and *Figure 2—figure supplement 1C*, *Figure 2—figure supplement 2*, *Figure 5—figure supplement 1*, and *Figure 7—figure supplement 1*.

### Major datasets

The following dataset was generated:

| Author(s) | Year | Dataset title | Dataset URL | Database, license, and accessibility information |
|---|---|---|---|---|
| Eichhorn SW, Subtelny AO, Kronja I, Kwasnieski JC, Orr-Weaver TL, Bartel DP | 2016 | mRNA poly(A)-tail changes specified by deadenylation broadly reshape translation in Drosophila oocytes and early embryos | https://www.ncbi.nlm.nih.gov/geo/query/acc.cgi?acc=GSE83616 | Publicly available at the NCBI Gene Expression Omnibus (accession no. GSE83616) |

The following previously published dataset was used:

| Author(s) | Year | Dataset title | Dataset URL | Database, license, and accessibility information |
|---|---|---|---|---|
| Kronja I, Eichhorn SW, Yuan B, Bartel DP, Orr-Weaver TL | 2014 | Polysome profiling and ribosome footprinting of Drosophila mature oocyte and activated egg | http://www.ncbi.nlm.nih.gov/geo/query/acc.cgi?acc=GSE52799 | Publicly available at the NCBI Gene Expression Omnibus (accession no. GSE52799) |

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
