## [Decision Letter]

Thank you for submitting your article "mRNA Poly(A)-tail Changes Specified by Deadenylation Broadly Reshape Translation in *Drosophila* Oocytes and Early Embryos" for consideration by *eLife*. Your article has been favorably evaluated by James Manley (Senior editor) and three reviewers, one of whom, Elisa Izaurralde (Reviewer #1), is a member of our Board of Reviewing Editors.

The reviewers have discussed the reviews with one another and the Reviewing Editor has drafted this decision to help you prepare a revised submission.

Summary:

There are several points raised by reviewers 2 and 3 that in fact I prefer not to summarize because this will involve losing some important information and relevant comments provide by experts in the field. Most of the points raised by the reviewers are related to the interpretation of the results and involve reanalysis of the data or the inclusion of additional information. These points can be addressed by the authors without further experimental work.

*Reviewer #1:*

This is a very interesting manuscript that provides a significant amount of novel information on poly(A) tail changes and translation efficiency in several stages of oogenesis and embryogenesis in *Drosophila*. This is the first study in which such a comprehensive analysis has been performed and thus provides highly relevant information and unexpected observations, which open new avenues of research. Therefore, the manuscript is ideal for publication in *eLife*.

*Reviewer #2:*

This manuscript presents a global analysis of mRNA translation, measured using ribosome profiling, and poly(A) tail length, measured using PAL-seq, during the early stages of *Drosophila* development, spanning three important developmental transitions: oocyte maturation, egg activation, and the maternal-to-zygotic transition. These data provide interesting insights into the correlation between mRNA translation and poly(A) tail length during various stages of development, and represent a valuable resource. However, a number of issues require clarification, and the manuscript needs to be improved through additional analyses, and made more useful to the reader through the presentation of additional information, as described in the specific comments below:

1) In comparing changes in poly(A) tail length, the authors largely rely on examination of fold change. However, while measurement of fold differences is one approach for quantifying poly(A) tail length changes, this approach is not sensitive and not the most informative when the reference stage gene has a long poly(A) tail. For example, a change in poly(A) length from 10 to 20 yields a 2-fold change; however, a poly(A) tail lengthened from 40 to 80 also yields a 2-fold change, despite the fact that the absolute change is four times greater for the second gene compared to the first gene. This absolute change is not reflected in examining the fold change of the tail length, which may be problematic if one assumes that PABP has a role in linking changes in tail length to changes in translation. The minimum poly(A) length bound by PABP is 12 residues and, on longer tracts, PABP binds approximately every 25 residues (Mol Cell Biol. 1987 Sep;7(9):3268, RNA 2005 Jul;11(7):1131). PABP binding-site length and the additional number of possible PABP copies bound (i.e. change in length as measured by the number of additional PABP that can bind to the transcript) need to be considered when examining changes in poly(A) tail length, and when performing analysis of TE. This analysis would be particularly enlightening in explaining large changes in TE that correspond to apparently small fold-changes in poly(A) tail length.

2) The section discussing the wispy mutant experiments is difficult to follow, and has a number of potential caveats. For instance:

A) The use of poly(A) selection in these experiments complicates the interpretation of the results; why would poly(A) selection in generating samples for RNAseq show biases in *wispy* mutants but not wild type?

B) If *wispy* mutants are blocked at the OET, how were TE and tail lengths measured in 0-2 h embryos (this isn't mentioned until the Discussion section)?

C) In Figure 5, while the changes in TE in *wispy* mutant and wild type during the OET are correlated, those in the *wispy* mutant appear decreased in magnitude, implying that *wispy* does have a role in regulating translational activation during this transition.

Given these issues, the conclusions from this analysis regarding a lack of dependence of translational activation on cytoplasmic polyadenylation are overstated.

3) The authors should provide lists and further analysis (e.g. GO-term enrichment) of specific groups of genes that are mentioned in the manuscript. For example, 300 transcripts underwent little to no decrease in TE despite substantial tail-length reductions during oocyte maturation; mRNAs with tails that were least affected by the absence of Wispy during OET; genes listed in Figure 5 and *png*-dependent up and down-regulated TE genes list in Figure 6.

4) mRNA targets of SMG have been defined with regard to SMG-associated transcripts, and its regulation of translation and decay (Chen et al., 2014; Genome Biology). How do transcripts up-regulated in *smg* mutants in this study compare to those previously identified as SMG targets?

5) Quantification of several parameters are needed when making comparative statements. For example, in the second paragraph of the subsection “A conserved switch in the nature of translational control”, the manuscript describes the slope of several relationships and comments on how the slope becomes "strongly diminished," but fails to provide numerical values of the slope, and thus makes it difficult to quantitatively judge these statements.

6) To maximize the usefulness of the manuscript to readers, it will be important to include the TE and poly(A) tail length data at different stages and in different genotypes as supplemental data. Authors should have information tables for different stage samples as in Kronja et al. (2014b), Table S2: showing each gene's mRNA abundance (rpkm), ribosome protected fragments (rpkm), translational efficiency, and mean or median of poly(A) tail length for wild type and each mutant genotype. In addition, links to the data on GEO should be provided.

7) The authors should provide more representative examples of the behaviour of individual transcripts from the datasets they are describing; for instance, in Figure 1 it would be of interest to see traces for additional key genes, such as *png, smg, bcd* and *wisp*.

8) Could the authors extend Figure 3 to include time points all the way up to embryo 2-3h? Alternatively, provide a similar analysis for embryonic stages? It would also be beneficial for the reader if each cluster shown in Figure 3 had graphs describing not only TE changes but poly(A) length changes as well, as in Figure 3. This would help demonstrate the relationship between TE and poly(A) tail length for the different clusters of transcripts.

9) Figure 6 and Figure 7 should be consistent (especially regarding the x- and y-axes) so that the reader can more easily compare the changes in *png*-dependent mRNAs in the *png* mutant to the changes in the *smg* mutant.

10) This manuscript includes both newly acquired and previously published data (Kronja et al., 2014a,b). The authors should more clearly indicate which data have been previously published and which are new. For example, a table should be provided outlining the source of the datasets used for each portion of this study: the table should indicate where new datasets or previously published datasets are used, and the type of sequence analysis performed (RNA-seq, ribosome-profile, and/or PAL-seq).

11) When describing the data in Figure 3, the authors discuss how a vast majority of the mRNAs that have >2-fold increase in TE are observed to undergo increases in poly(A) tail length. However, without the reciprocal analysis, this approach appears biased towards confirming their current model. The authors should perform the reciprocal analysis and examine the TE behavior of the mRNAs that undergo poly(A) tail lengthening, and show what proportion of the lengthened mRNAs also have a >2-fold increase in TE.

12) More information should be provided in the Methods and/or main text with regard to the number of replicates carried out for different experiments and statistical analyses performed on the data.

*Reviewer #3:*

The work by Eichhorn et al. characterizes the correlation between poly(A) tail length changes and ribosome-footprinting/mRNAs-seq (Translation Eficiency; TE) in *Drosophila* oocytes and embryos. Thus, the authors describe the correlation, or lack of correlation, between both events at different developmental stages and in mutant flies for *wispy, png* and *smg* (all previously implicated in translation regulation and/or poly(A) tail length dynamics). This study is a direct follow-up on previous works by the authors studying changes in mRNA translation during early *Drosophila* development (Kronja et al., 2014, ribosome-footprinting and proteomics) and poly(A) tail length profiling, together with TE measurements, in zebrafish and frog early development (Subtelny, 2014). Conclusions of the current study confirm those of previous works, showing a strong correlation between poly(A) tail length changes and TE until gastrulation, when the coupling is dampened.

Now, the authors further extend these approaches by using mutant embryos for known regulators of embryonic mRNA translation. However, genome-wide correlations allow for a limited set of conclusions as to the mechanism of maternal mRNA translation regulation, being the added value more as "valuable resources for future studies".

1) The "aggregation" of data from different mRNA variants in PAL-Seq and TE complicates interpretation of results. Thus, in oocytes, where there is no transcription and where there is a good correlation between poly(A) tail length and TE, it is safe to assume that the same mRNA is being compared. However, after MZT, newly transcribed mRNAs (for any given ORF) can be different than the maternal ones. This is indeed the case for alternative cleavage and polyadenylation during embryonic development (Hoque, 2013).

2) This work assumes a linear relationship between poly(A) tail length and translation. However, this has not been demonstrated. For example, while previous works show that (for maternal mRNAs) elongation of the poly(A) tail from 20-30 to 80-100 As causes translational activation (presumably by allowing the formation of the close-loop eIF4E-eIF4G-PABP), there is no evidence showing that further elongation has any impact. Thus, more than a linear correlation, poly(A) tail length may have a bimodal regulatory effect. This is supported by recent evidence (Park et al., 2016) indicating that a single PABP (binding 30-40 As) may be sufficient to support full translational activation. Therefore, ΔP(A) does not have the same meaning when it goes from 20 to 40 As than from 100 to 200 As.

3) Even for "unimodal" poly(A) tail length distribution (i.e., *cyclin B*, Figure 4) the peak is broad ranging from 0 to 100, with about 40% of the transcripts having a poly(A) tail above 50 nt. Based on the above argument, it is difficult to interpret the meaning of an average at 40 nt. On the other hand, for toll (4D), most of the mRNAs are below 50 nt in stage 14 and the majority above 50 in activated eggs.

4) Normalization of the ribosome-footprinting by mRNA levels (mRNA-seq) using oligo-dT capture generates a strong bias that overestimates TE for short polyadenylated mRNAS (Park et al., 2016).

5) Being mRNA translation initiation a "competitive event" in which, not only mRNA-intrinsic features, but also competition with other mRNAs for the translational machinery dictates its efficiency, the large level changes in specific mRNAs after MZT (with maternal mRNA degradation and new transcription) can severely affect the normalization to obtain the TE ratio. This would not be an issue before MZT, as individual mRNA levels do not change.

6) Other level at which the interpretation of the results is difficult is derived from the origin of changes in poly(A) tail length, nuclear vs. cytoplasmic. Thus, in oocytes (with no transcription and no mRNA degradation), poly(A) tail length changes are presumably originated in the cytoplasm and for the same mRNA populations. After MZT (when mRNA transcription and degradation are reestablished), most of the poly(A) tail elongation events presumably correspond to newly transcribed mRNAs and most of the deadenylation events will result in mRNA degradation.

7) Probably, the most relevant contribution of this work over previous analyses is the use of mutants to correlate changes in poly(A) tail length with changes in TE. Although some potential correlations are found for *wispy, png* and *smg*, the main problem of this approach is that it does not differentiate between direct and indirect effects and all three genes have a profound impact in early embryonic development. This is clearly shown in the global deadenylation in *wispy*-mut oocytes and embryos. Obviously, this phenotype cannot be ascribed to the direct role of wispy in cytoplasmic polyadenylation, which should only affect a reduced number of mRNAs. In turn, the fact that these global changes do not impact the TE/poly(A) distribution over WT, can be due to the competitive nature of the poly(A) tail effect.

---

## [Author Response]

*Summary:*

*There are several points raised by reviewers 2 and 3 that in fact I prefer not to summarize because this will involve losing some important information and relevant comments provide by experts in the field. Most of the points raised by the reviewers are related to the interpretation of the results and involve reanalysis of the data or the inclusion of additional information. These points can be addressed by the authors without further experimental work.*

*Reviewer #2:*

*This manuscript presents a global analysis of mRNA translation, measured using ribosome profiling, and poly(A) tail length, measured using PAL-seq, during the early stages of Drosophila development, spanning three important developmental transitions: oocyte maturation, egg activation, and the maternal-to-zygotic transition. These data provide interesting insights into the correlation between mRNA translation and poly(A) tail length during various stages of development, and represent a valuable resource. However, a number of issues require clarification, and the manuscript needs to be improved through additional analyses, and made more useful to the reader through the presentation of additional information, as described in the specific comments below:*

*1) In comparing changes in poly(A) tail length, the authors largely rely on examination of fold change. However, while measurement of fold differences is one approach for quantifying poly(A) tail length changes, this approach is not sensitive and not the most informative when the reference stage gene has a long poly(A) tail. For example, a change in poly(A) length from 10 to 20 yields a 2-fold change; however, a poly(A) tail lengthened from 40 to 80 also yields a 2-fold change, despite the fact that the absolute change is four times greater for the second gene compared to the first gene. This absolute change is not reflected in examining the fold change of the tail length, which may be problematic if one assumes that PABP has a role in linking changes in tail length to changes in translation. The minimum poly(A) length bound by PABP is 12 residues and, on longer tracts, PABP binds approximately every 25 residues (Mol Cell Biol. 1987 Sep;7(9):3268, RNA 2005 Jul;11(7):1131). PABP binding-site length and the additional number of possible PABP copies bound (i.e. change in length as measured by the number of additional PABP that can bind to the transcript) need to be considered when examining changes in poly(A) tail length, and when performing analysis of TE. This analysis would be particularly enlightening in explaining large changes in TE that correspond to apparently small fold-changes in poly(A) tail length.*

Inspired by the referee’s comment, we repeated our analyses of the relationship between changes in tail length and changes in TE throughout development (i.e., those appearing in Figure 1, Figure 2, Figure 4, Figure 5, Figure 6 and Figure 7) using absolute changes in tail length instead of fold differences (log_2_ fold changes). This analysis resulted in plots that were essentially identical to those made using log_2_ fold changes, and in all cases the correlation between change in tail length and change in TE was essentially the same as when it was calculated using log_2_ fold changes in tail length. These new analyses are now included as Figure 7—figure supplement 2.

Part of the reason that these two methods of analyzing the impact of changes in tail length on TE yielded such similar results is because the absolute change in tail length is highly correlated with the log_2_ fold change. However, we, like the referee, might have expected the absolute tail-length changes to still be more informative if the main influence of changing the tail was to change the number of PABP proteins that could bind. If, however, PABP was limiting during the stages in which we observed coupling between tail length and TE changes, and mRNAs were competing for the limiting pool of PABP, then the relative changes would be at least as informative as absolute changes because a 2-fold lengthening would be expected to have the same relative effect on ability to compete for PABP regardless of whether a poly(A)-tail increased from 20 to 40 nt or from 40 to 80 nt. Thus, our observation that relative changes were as informative as absolute changes implies that mRNAs are competing for PABP. We have revised our text to include this mechanistic implication of our new analyses.

2) The section discussing the wispy mutant experiments is difficult to follow, and has a number of potential caveats. For instance:

*A) The use of poly(A) selection in these experiments complicates the interpretation of the results; why would poly(A) selection in generating samples for RNAseq show biases in wispy mutants but not wild type?*

In the *wispy*-mutant samples, the poly(A)-tail length of essentially every gene was shorter than in the corresponding wild-type samples (Figure 5). Due to the global shortening of tails in the *wispy*-mutant samples, many of the tail lengths now fell within a size range that was not efficiently captured during poly(A)- selection. Moreover, poly(A)-tail length and mRNA abundance correlated in the *wispy*-mutant samples but not in the wild-type samples (Figure 5—figure supplement 1), again supporting the idea that mRNAs with short tails were poorly recovered in the *wispy*-mutant samples, but that tail lengths were sufficiently long for this problem to not arise in the wild-type samples. Additional analysis discussed in the paper pertaining to [Supplementary-material SD1-data] supports the absence of a poly(A)-selection bias in our wild-type samples, indicating that this was only an issue in the *wispy*-mutant due to the pervasive tail-shortening that occurred. We have revised the discussion of the *wispy*-mutant experiments to make it easier to follow.

*B) If wispy mutants are blocked at the OET, how were TE and tail lengths measured in 0-2 h embryos (this isn't mentioned until the Discussion section)?*

Although *wispy*-mutant eggs do not complete meiosis and thus do not begin nuclear divisions as embryos would, most of the molecular changes characteristic of the OET are independent of these divisions, motivating a comparison of *wispy*-mutant laid eggs with wild-type early embryos. We now explain this in the Results and refer to *wispy*-mutant samples at this stage as “laid eggs” (rather than “0–2 h embryos”) to help clarify this point.

*C) In Figure 5, while the changes in TE in wispy mutant and wild type during the OET are correlated, those in the wispy mutant appear decreased in magnitude, implying that wispy does have a role in regulating translational activation during this transition.*

Overall, the changes in the *wispy* mutant do not decrease in magnitude. A linear deming regression of these data in Figure 5 yields a slope of 1.2 (red line in Figure 8), which is consistent with *wispy*-mutant data exhibiting a slightly largerchange in TE than we observed in the corresponding wild-type data.

Author response image 1.**DOI:**
http://dx.doi.org/10.7554/eLife.16955.019

*Given these issues, the conclusions from this analysis regarding a lack of dependence of translational activation on cytoplasmic polyadenylation are overstated.*

We have continued to analyze the *wispy* data, considering even those mRNAs without any detected translation in the *wispy* mutant (Figure 5—figure supplement 1), which has provided some evidence for *wispy*-dependent translational activation, and we have tempered our conclusions accordingly.

*3) The authors should provide lists and further analysis (e.g. GO-term enrichment) of specific groups of genes that are mentioned in the manuscript. For example, 300 transcripts underwent little to no decrease in TE despite substantial tail-length reductions during oocyte maturation; mRNAs with tails that were least affected by the absence of Wispy during OET; genes listed in Figure 5 and png-dependent up and down-regulated TE genes list in Figure 6.*

We have provided the requested lists of specific groups of genes mentioned in the manuscript ([Supplementary-material SD3-data]). Throughout the text we have highlighted classes of genes that were over-represented within a group of genes with a characteristic behavior. For example, we mention and discuss the proteasome genes as highly enriched among the ~300 transcripts that decrease in poly(A)-tail length without concordant TE decreases during oocyte maturation, and mention the ribosomal protein genes as highly enriched among the mRNAs with tails least affected by loss of *wispy*. The requested GO-term enrichment analysis for *png*-dependent up and down-regulated genes appears in a prior manuscript cited when describing these sets of targets (Kronja et al., 2014b). We have also performed many additional analyses of GO-term enrichment that are not presented in the manuscript, choosing instead to focus on the most significant results and highlight only those groups of genes that seem biologically relevant or particularly striking. Anyone interested in the weaker, less interesting GO-term enrichments will be able to easily obtain these enrichments themselves using the processed data from our [Supplementary-material SD2-data] (point 6, below).

*4) mRNA targets of SMG have been defined with regard to SMG-associated transcripts, and its regulation of translation and decay (Chen et al., 2014; Genome Biology). How do transcripts up-regulated in smg mutants in this study compare to those previously identified as SMG targets?*

As requested, we analyzed these previously identified SMG-bound mRNAs and found the anticipated decreases in TE and tail length in wild-type samples but not in the *smg*-mutant samples. We have added this analysis as Figure 7—figure supplement 1.

*5) Quantification of several parameters are needed when making comparative statements. For example, in the second paragraph of the subsection “A conserved switch in the nature of translational control”, the manuscript describes the slope of several relationships and comments on how the slope becomes "strongly diminished," but fails to provide numerical values of the slope, and thus makes it difficult to quantitatively judge these statements.*

Throughout the paper these statements have either been replaced or expanded upon with quantitative metrics. For example, the statement that the referee mentions now says, “the slope of this relationship strongly diminished such that the median TE of mRNAs with a poly(A)-tail length between 70–80 nt was <2-fold greater than that of mRNAs with a poly(A)-tail length between 30-40 nt.”

*6) To maximize the usefulness of the manuscript to readers, it will be important to include the TE and poly(A) tail length data at different stages and in different genotypes as supplemental data. Authors should have information tables for different stage samples as in Kronja et al. (2014b), Table S2: showing each gene's mRNA abundance (rpkm), ribosome protected fragments (rpkm), translational efficiency, and mean or median of poly(A) tail length for wild type and each mutant genotype. In addition, links to the data on GEO should be provided.*

As requested, all of the RNA, RPF, TE, and poly(A)-tail length measurements now appear in [Supplementary-material SD2-data], with each developmental stage as its own sheet within the excel workbook, and information about whether a given gene passed the cutoffs applied throughout our analyses to aid readers in reproducing our results. Additionally, these processed files as well as the raw data will be available for download from GEO (accession numbers GSE52799 and GSE83616).

*7) The authors should provide more representative examples of the behaviour of individual transcripts from the datasets they are describing; for instance, in Figure 1 it would be of interest to see traces for additional key genes, such as png, smg, bcd and wisp.*

Figure 1 has been updated to include *smaug* and *bicoid*. We have not included the same RNA/TE traces for *png* and *wisp* because neither seems to be controlled through translational upregulation, and thus their inclusion would make the point of this figure harder to understand. We have included *png* and *wisp* traces in Figure 9, which imply that their upregulation is post-translational and that they are ultimately repressed through mRNA destabilization.

Author response image 2.**DOI:**
http://dx.doi.org/10.7554/eLife.16955.020

*8) Could the authors extend Figure 3 to include time points all the way up to embryo 2-3h? Alternatively, provide a similar analysis for embryonic stages? It would also be beneficial for the reader if each cluster shown in Figure 3 had graphs describing not only TE changes but poly(A) length changes as well, as in Figure 3. This would help demonstrate the relationship between TE and poly(A) tail length for the different clusters of transcripts.*

We have added the requested analysis for embryonic stages (from stage 14 of oogenesis through 2-3 h of embryonic development) as Figure 1—figure supplement 1.

Our revised analysis resolved the discordant behavior we previously saw in the relationship between tail-length changes and TE changes for up and down- regulated mRNAs (see revised text and Figure 4). For this reason we have removed the previous Figure 3 and corresponding text and believe the revised Figure 3 and Figure 4 are sufficient for demonstrating the now simpler relationship between tail length and TE changes.

*9) Figure 6 and Figure 7 should be consistent (especially regarding the x- and y-axes) so that the reader can more easily compare the changes in png-dependent mRNAs in the png mutant to the changes in the smg mutant.*

We have made Figure 6 and Figure 7 consistent, although we note that in Figure 6, activated egg is being compared to stage 14, whereas in Figure 7, 0–1 h embryo is being compared to stage 14, and for this reason the data should not be directly compared.

*10) This manuscript includes both newly acquired and previously published data (Kronja et al., 2014a,b). The authors should more clearly indicate which data have been previously published and which are new. For example, a table should be provided outlining the source of the datasets used for each portion of this study: the table should indicate where new datasets or previously published datasets are used, and the type of sequence analysis performed (RNA-seq, ribosome-profile, and/or PAL-seq).*

The RNA-seq and RPF data for wild-type stage 14 oocytes and activated eggs and *png*-mutant activated eggs that are analyzed in this paper were previously published in Kronja et al., 2014b, and this is now more clearly described in the Methods section and stated in all figures where these data were used. We note that the vast majority of our datasets, including the poly(A)-profiling datasets from the same samples as the previously published RNA-seq and RPF datasets, have not been previously published.

*11) When describing the data in Figure 3, the authors discuss how a vast majority of the mRNAs that have >2-fold increase in TE are observed to undergo increases in poly(A) tail length. However, without the reciprocal analysis, this approach appears biased towards confirming their current model. The authors should perform the reciprocal analysis and examine the TE behavior of the mRNAs that undergo poly(A) tail lengthening, and show what proportion of the lengthened mRNAs also have a >2-fold increase in TE.*

As mentioned in our response to point 8, our revised analysis resolved the discordant behavior we previously observed in the relationship between tail length changes and TE changes for up and down-regulated mRNAs during oocyte maturation. This behavior occurred because we had erroneously used TE data and PAL-seq data that came from two different stage 14 samples. When we used measurements made from the same sample, this discordant behavior went away, and thus the analysis that the referee would like for us to improve no longer appears in the paper.

*12) More information should be provided in the Methods and/or main text with regard to the number of replicates carried out for different experiments and statistical analyses performed on the data.*

We have expanded the Methods section to include a paragraph about reproducibility and the statistical analyses performed on the data.

*Reviewer #3:*

*The work by Eichhorn et al. characterizes the correlation between poly(A) tail length changes and ribosome-footprinting/mRNAs-seq (Translation Eficiency; TE) in Drosophila oocytes and embryos. Thus, the authors describe the correlation, or lack of correlation, between both events at different developmental stages and in mutant flies for wispy, png and smg (all previously implicated in translation regulation and/or poly(A) tail length dynamics). This study is a direct follow-up on previous works by the authors studying changes in mRNA translation during early Drosophila development (Kronja et al., 2014, ribosome-footprinting and proteomics) and poly(A) tail length profiling, together with TE measurements, in zebrafish and frog early development (Subtelny, 2014). Conclusions of the current study confirm those of previous works, showing a strong correlation between poly(A) tail length changes and TE until gastrulation, when the coupling is dampened.*

*Now, the authors further extend these approaches by using mutant embryos for known regulators of embryonic mRNA translation. However, genome-wide correlations allow for a limited set of conclusions as to the mechanism of maternal mRNA translation regulation, being the added value more as "valuable resources for future studies".*

The referee seems to imply that genome-wide correlations are of limited value with regard mechanistic insight and that results that provide mechanistic insight are the only type of results that offer value. We have a more positive view of what can be learned from high-throughput analyses and what constitutes a valuable result. Indeed, from our analyses, we made the following discoveries and insights:

A) *Drosophila* has a developmental shift in the nature of translational control, in which the coupling between translational efficiency and poly(A)-tail length disappears at gastrulation. This developmental shift resembles the one that had previously been observed only in vertebrates—the fact that it is also observed in an invertebrate implies that it is a basal feature of bilaterian development and post-transcriptional gene regulation.

B) Widespread coupling between translational efficiency and tail length extends to oocyte development. Previously it had been observed globally only during embryonic development, or only for a few genes in oocytes.

C) Key developmental regulators, such as *zelda, stat92E, bicoid, cyclin B, fizzy*, and *torso*, which were previously thought to be activated by tail-length– dependent processes, are also activated through other mechanisms.

D) The extent to which tail-length changes can explain the translational regulation by PAN GU and Smaug is greater than previously appreciated.

E) In *Drosophila*, the differential translational efficiencies of different mRNAs observed in late oocytes and early embryos are specified more by selective deadenylation than by selective cytoplasmic polyadenylation. This result differs from the paradigm that operates in vertebrate oocytes/embryos and had been assumed to be operating in *Drosophila*.

Thus our work not only provides a resources for future studies but also new discoveries, important in their own right.

*1) The "aggregation" of data from different mRNA variants in PAL-Seq and TE complicates interpretation of results. Thus, in oocytes, where there is no transcription and where there is a good correlation between poly(A) tail length and TE, it is safe to assume that the same mRNA is being compared. However, after MZT, newly transcribed mRNAs (for any given ORF) can be different than the maternal ones. This is indeed the case for alternative cleavage and polyadenylation during embryonic development (Hoque, 2013).*

Nearly all analyses presented in this paper are on samples that are not transcriptionally active (the exception being the right half of Figure 1, Figure 1—figure supplement 1, and three scatter plots of Figure 7—figure supplement 2). Indeed, all but the first of the five discoveries/insights listed above were made based on the analysis of pre-MZT embryos. Additionally, our analyses of zebrafish and *Xenopus* (Subtelny et al., 2014), found that coupling between poly(A)-tail length and TE extended well past the onset of transcription (zygotic transcription begins at approximately 2.5 hours post-fertilization in zebrafish and stage 7 in *Xenopus*, whereas we observed coupling between tail length and TE at 4 hours post-fertilization and stage 9, respectively). Thus, there is no reason to think that the loss of coupling we observe around gastrulation in zebrafish, *Xenopus*, and *Drosophila* resulted from any potential issues relating to different transcript isoforms.

*2) This work assumes a linear relationship between poly(A) tail length and translation. However, this has not been demonstrated. For example, while previous works show that (for maternal mRNAs) elongation of the poly(A) tail from 20-30 to 80-100 As causes translational activation (presumably by allowing the formation of the close-loop eIF4E-eIF4G-PABP), there is no evidence showing that further elongation has any impact. Thus, more than a linear correlation, poly(A) tail length may have a bimodal regulatory effect. This is supported by recent evidence (Park et al., 2016) indicating that a single PABP (binding 30-40 As) may be sufficient to support full translational activation. Therefore, ΔP(A) does not have the same meaning when it goes from 20 to 40 As than from 100 to 200 As.*

The single-PABP result of Park et al. is not relevant to the developmental stages for which we see coupling between translational efficiency and tail length, since this observation was made in a very different regulatory regime (i.e., the typical post-embryonic regime for which we had already shown that translational efficiency does not generally correlate with tail length; Subtelny et al., 2014).

Nonetheless, the idea that translational activation might plateau after tail elongation exceeds 100 nt is potentially pertinent to our study, since it is based on observations of maternal mRNAs, implying that these observations were made in a developmental stage at which we have shown that translational efficiency is coupled to tail length (Figure 1 and Figure 2, and Subtelny et al., 2014). Indeed, we sometimes did observe a plateauing of the relationship between tail length and translation in our analyses (e.g., Figure 2, stage 11, 12, and 13 oocytes), which resembled the non-linear relationship that the referee mentions. Interestingly, though, we did not observe a plateau when comparing *changes* in tail length with changesin translational efficiency in the two developmental transitions that retain coupling (oocyte maturation and egg activation) (Figure 4). Moreover, this plateau was not a major factor in dampening the relationship between translational efficiency and tail length, as very few mRNAs had mean tail lengths exceeding 100 nt in our *Drosophila* samples.

With regard to our work assuming a linear relationship, we have always used Spearman analysis of correlations (which avoids the linear assumption), but our initial submission also did include some linear regression analyses of the relationship between tail length and translation, which indeed implied that we were assuming a linear relationship between the two. In the revision, we have replaced the previous regression analysis with an analysis of median differences in translation for mRNAs with mean tail lengths of 30–40 nt compared to those with mean tail lengths of 70–80 nt (within the range that the referee proposes to be influential for maternal mRNAs). Therefore, we no longer imply a linear relationship between tail length and translation, and none of our conclusions depend on the assumption of a linear relationship.

Finally, we note that this concern of referee #3, that doubling of a short tail would be more consequential than doubling of a long tail, resembles the concern of referee #2 (point 1) but is in the opposite direction, as referee #2 is concerned that doubling of the long tail would be more consequential than doubling of the short tail. Thus, had we found that using absolute tail-length changes (rather than relative tail length changes) reduced the correlations with TE, we would have had support for the concern of referee #3, but this was not the case (see response to referee #2, point 1, and Figure 7—figure supplement 2 of the revised manuscript).

*3) Even for "unimodal" poly(A) tail length distribution (i.e., cyclin B, Figure 4) the peak is broad ranging from 0 to 100, with about 40% of the transcripts having a poly(A) tail above 50 nt. Based on the above argument, it is difficult to interpret the meaning of an average at 40 nt. On the other hand, for toll (4D), most of the mRNAs are below 50 nt in stage 14 and the majority above 50 in activated eggs.*

There is indeed a distribution of poly(A)-tail lengths for mRNAs from each gene. We have tried considering the entire distribution of tail lengths for each mRNA and treating the relationship with tail length as a step function, trying an array of parameters, but none of these more complicated treatments were more informative than using the simple mean. The strong performance of the mean was presumably because the tail-length distributions are unimodal and thus an mRNA with a longer mean tail length has more longer-tail molecules than one with a shorter mean tail length, and when the mean tail length increases, more molecules have longer tails. Therefore, even if the relationship involves a step function, our simple approach of using the mean captures enough of the relevant information to perform as well as more complicated treatments.

*4) Normalization of the ribosome-footprinting by mRNA levels (mRNA-seq) using oligo-dT capture generates a strong bias that overestimates TE for short polyadenylated mRNAS (Park et al., 2016).*

We have also been mindful of the potential pitfalls of using oligo-dT capture for mRNA-seq used for normalization of ribosome-footprinting experiments (Weinberg et al., Cell Reports2016). To investigate the possibility that this influenced the TE measurements for our *Drosophila* samples, we compared RNA expression levels between all of our samples. In oocytes from stages 11 through 14, activated eggs, and 0-1h embryos, transcription and mRNA degradation are effectively absent, but many changes in poly(A)-tail length occurred. These changes in poly(A)-tail length did not result in changes in mRNA abundance ([Supplementary-material SD1-data]), which showed that in wild-type *Drosophila* samples poly(A)-tail lengths were long enough that oligo-dT selection did not introduce a bias. In contrast, we did observe a bias resulting from oligo-dT selection in the *wispy*- mutant samples (Figure 5—figure supplement 1), which had poly(A)-tail lengths that were much shorter than those in the corresponding wild-type samples. In the text we describe how we account for this bias in our analyses of these data.

*5) Being mRNA translation initiation a "competitive event" in which, not only mRNA-intrinsic features, but also competition with other mRNAs for the translational machinery dictates its efficiency, the large level changes in specific mRNAs after MZT (with maternal mRNA degradation and new transcription) can severely affect the normalization to obtain the TE ratio. This would not be an issue before MZT, as individual mRNA levels do not change.*

As noted in our response to point 1, essentially all the analyses presented in the paper deal with pre-MZT samples (the exceptions being the right half of Figure 1, Figure 1—figure supplement 1, and three scatter plots of Figure 7—figure supplement 2).

Following the MZT, changes in the set of expressed transcripts could indeed influence the TE value of a given gene. Our revised manuscript emphasizes that TE is a relative measure and should be interpreted as reflecting how well an mRNA is translated relative to the other mRNAs present in a given context, and that changes in the translation of the other mRNAs could cause a change in TE for an mRNA that is actually translated at the same efficiency (subsection “A conserved switch in the nature of translational control”, fifth paragraph).

*6) Other level at which the interpretation of the results is difficult is derived from the origin of changes in poly(A) tail length, nuclear vs. cytoplasmc. Thus, in oocytes (with no transcription and no mRNA degradation), poly(A) tail length changes are presumably originated in the cytoplasm and for the same mRNA populations. After MZT (when mRNA transcription and degradation are reestablished), most of the poly(A) tail elongation events presumably correspond to newly transcribed mRNAs and most of the deadenylation events will result in mRNA degradation.*

This is indeed our interpretation of the changes in poly(A)-tail length that we see before and after MZT, as now explained in the subsection “A conserved switch in the nature of translational control”. As with previous points, it is important to keep in mind that nearly all of our analyses are on the more simply interpreted pre-MZT samples.

*7) Probably, the most relevant contribution of this work over previous analyses is the use of mutants to correlate changes in poly(A) tail length with changes in TE. Although some potential correlations are found for wispy, png and smg, the main problem of this approach is that it does not differentiate between direct and indirect effects and all three genes have a profound impact in early embryonic development. This is clearly shown in the global deadenylation in wispy-mut oocytes and embryos. Obviously, this phenotype cannot be ascribed to the direct role of wispy in cytoplasmic polyadenylation, which should only affect a reduced number of mRNAs. In turn, the fact that these global changes do not impact the TE/poly(A) distribution over WT, can be due to the competitive nature of the poly(A) tail effect.*

Because Wispy is the only cytoplasmic poly(A) polymerase acting at the stages examined, we in fact can ascribe the molecular phenotype of the *wispy* mutant (i.e., global deadenylation) to the direct role of Wispy in cytoplasmic polyadenylation. When we combine this finding with our observation that this global tail shortening only modestly impacts the TE/poly(A) distribution over WT, we arrive at one of the major conclusions of our study (listed as #5 in the first part of our response to this referee), that in contrast to the prevailing paradigm, the differential translational efficiencies of different mRNAs observed in *Drosophila* oocytes and early embryos are specified more by selective deadenylation than by selective cytoplasmic polyadenylation. The competitive nature of translation and the poly(A) effect does indeed explain why the TE distribution still reflects the poly(A) tail distribution in the *wispy* mutant, but the key finding is that relative differences in tail lengths are largely maintained in the absence of selective cytoplasmic polyadenylation.

With respect to PNG and SMG, indirect effects certainly occur, given that these regulators act on large sets of mRNAs. Nonetheless, our analysis of the *smg*- mutant data found that PNG-dependent downregulated mRNAs were translationally derepressed, while the *png*-dependent upregulated mRNAs were not affected by loss of SMG. This result shows that indirect effects did not overwhelm the signal of direct effects in this experiment. Moreover, our revision also includes an analysis of a set of mRNAs known to bind SMG, which should be enriched in direct targets (Figure 7—figure supplement 1).